# The Ocean’s Pharmacy: Health Discoveries in Marine Algae

**DOI:** 10.3390/molecules29081900

**Published:** 2024-04-22

**Authors:** Mélanie Silva, Dorit Avni, João Varela, Luísa Barreira

**Affiliations:** 1Centre of Marine Sciences, University of Algarve, 8005-139 Faro, Portugal; mvsilva@ualg.pt (M.S.); jvarela@ualg.pt (J.V.); 2MIGAL Galilee Institute, Kiryat Shmona 1106000, Israel; dorita@migal.org.il; 3Green Colab—Associação Oceano Verde, University of Algarve, 8005-139 Faro, Portugal

**Keywords:** health, bioactivity, algae, microalgae, macroalgae, non-communicable diseases, in vivo studies, in vitro studies, cell assays

## Abstract

Non-communicable diseases (NCDs) represent a global health challenge, constituting a major cause of mortality and disease burden in the 21st century. Addressing the prevention and management of NCDs is crucial for improving global public health, emphasizing the need for comprehensive strategies, early interventions, and innovative therapeutic approaches to mitigate their far-reaching consequences. Marine organisms, mainly algae, produce diverse marine natural products with significant therapeutic potential. Harnessing the largely untapped potential of algae could revolutionize drug development and contribute to combating NCDs, marking a crucial step toward natural and targeted therapeutic approaches. This review examines bioactive extracts, compounds, and commercial products derived from macro- and microalgae, exploring their protective properties against oxidative stress, inflammation, cardiovascular, gastrointestinal, metabolic diseases, and cancer across in vitro, cell-based, in vivo, and clinical studies. Most research focuses on macroalgae, demonstrating antioxidant, anti-inflammatory, cardioprotective, gut health modulation, metabolic health promotion, and anti-cancer effects. Microalgae products also exhibit anti-inflammatory, cardioprotective, and anti-cancer properties. Although studies mainly investigated extracts and fractions, isolated compounds from algae have also been explored. Notably, polysaccharides, phlorotannins, carotenoids, and terpenes emerge as prominent compounds, collectively representing 42.4% of the investigated compounds.

## 1. Introduction

Over the course of four billion years and since the first life form, marine life has evolved considerably. As a result, this primordial ecosystem [1,2], which is essential to life on Earth, has presented a high level of biodiversity throughout time, offering an abundance of potential resources that are unique to this environment, thus acting as a valuable repository of new bioactive compounds, presenting promising opportunities for the discovery of drugs with unparalleled chemical novelty [3,4]. The exploitation of marine natural products (MNPs) is relatively recent (1950s) [4], whereas the scientific community’s interest in their hidden potential is constantly rising, and, to date, over 30,000 MNPs have been uncovered. Yet, these sources, including algae, are still considered to be largely untapped [1,5].

Algae, which can be multicellular (seaweed or macroalgae), unicellular, or colonial (microalgae, including cyanobacteria) organisms, are extraordinary reservoirs of biodiversity, with more than 30,000 identified species. However, even the most conservative estimations state this number as being at least as high as the undiscovered part [6]. Macroalgae can be classified into three phyla, based on their pigmentation: Chlorophyta (green algae) such as *Ulva* and *Codium*; Rhodophyta (red algae) such as *Chondrus* and *Pyropia*; and Ochrophyta (brown algae) such as *Ecklonia* and *Saccharina* [7]. Microalgae can be subdivided into Cyanophyta (blue-green prokaryotic algae such as *Oscillatoria*), Chlorophyta (green eukaryotic algae such as *Chlorella*), Rhodophyta (red eukaryotic algae such as *Porphyridium*), Chrysophyta (golden eukaryotic diatoms as *Phaeodactylum*), and Pyrrophyta (brown eukaryotic dinoflagellates such as *Ceratium*) [8,9].

These organisms are mostly photoautotrophic and use carbon dioxide from the atmosphere or the marine environment as a carbon source and sunlight as an energy source through photosynthesis, producing oxygen and being considered sustainable feedstocks [10,11]. Additionally, algae have an interesting nutritional profile, being rich in essential nutrients, such as proteins, vitamins, minerals, carbohydrates (including fibers), or lipids (with a focus on mono- and polyunsaturated fatty acids), depending on species or cultivation methods, among other factors [11,12]. MNPs from algae include peptides, lectins, carotenoids (e.g., fucoxanthin and *β*-carotene), polysaccharides, enzymes, vitamins, fatty acids, phenolic compounds (e.g., flavonoids), and phytosterols [10,12]. These MNPs have been found to possess numerous bioactivities, including antimicrobial, neuroprotective, cytotoxic, anti-aging, aggregative, anti-diabetic, vasoconstricting, anti-fungal, anti-tumoral, hypocholesterolemia, antioxidant, anti-inflammatory, immunosuppressive, anti-fouling, and antiviral properties, which can help in the mitigation of many human health-related issues, including non-communicable diseases [10,12,13].

Non-communicable diseases (NCDs) are non-infectious chronic pathologies regarded as one of the significant health challenges in the 21st century, with some describing it as being this century’s epidemic. NCDs are the primary cause of mortality (accounting for 74% of all fatalities and for over 80% of premature deaths) and disease burden globally. NCDs can exacerbate the occurrence of other illnesses, leading to a further decline in the quality of life for those affected and resulting in preventable long-term incapacity among patients. The main NCDs are cardiovascular diseases (such as heart attacks and strokes), cancers, chronic respiratory diseases, and diabetes [14,15]. Obesity and overweight are metabolic risk factors that significantly enhance the likelihood of developing non-communicable diseases. These conditions can lead to gut dysbiosis, which has been well documented as a contributing factor to NCDs [16,17,18]. Oxidative stress and chronic inflammation also represent cornerstones in the development and progression of NCDs and are often targets for drug development [19,20].

A significant number of conventional therapeutic medications (e.g., orlistat, epirubicin, and acarbose) used to treat or manage NCDs exhibit considerable adverse effects (e.g., hepatotoxicity, cardiotoxicity, and gastrointestinal, neurologic, and renal disturbances) due to their low selectivity [21,22]. As a result, there is a growing interest in finding new, more natural, and targeted therapeutic approaches. Therefore, various novel leads for pharmaceuticals have been investigated, including those based on natural products [4,23]. According to the World Health Organization, natural products are the main source of therapeutic agents worldwide [1]. Although most of these known compounds come from land-based sources like plants and bacteria, there is a largely unexplored collection of marine natural products. Algae, in particular, hold great potential as a source of unique bioactive compounds with structures that could be valuable for drug research [4]. As a result, they can induce health benefits through multiple biological mechanisms, such as antioxidant, anti-inflammatory, cardioprotective, gut-health balancing, anti-adipogenic, and anti-cancerous activities, among others [4,24,25,26,27]. Consequently, algae possess the capability to offer valuable perspectives on distinctive chemical architectures for the purpose of drug research to aid in the treatment of non-communicable diseases [4].

This review aims to provide a comprehensive analysis of the potential pharmacological and biological uses of marine algae and/or their compounds by compiling existing knowledge and research findings. The primary goal of this review is to examine and assess the therapeutic potential of marine algae species, specifically their impact on human health and their capacity to generate novel pharmacological and health-promoting compounds for the treatment of human diseases and the enhancement of human well-being, aiming at six main properties or bioactivities: (1) antioxidant properties; (2) anti-inflammatory effects; (3) cardioprotective activity; (4) gastrointestinal health modulation; (5) metabolic-health-promoting activity; and (6) anti-cancer activity. This review will conclude by discussing future directions, identifying research gaps, and addressing issues associated with the use of algal products for drug development.

## 2. Methodology

This systematic review was performed according to the recommendations of the Preferred Reporting Items for Systematic Reviews and Meta-Analyses (PRISMA) statement.

The literature search was performed on the 25 September 2023 in two databases: Web of Science (https://www.webofscience.com/, accessed on 25 September 2023) and Wiley Online Library (https://onlinelibrary.wiley.com/, accessed on 25 September 2023). The used search string was as follows: (alga* OR seaweed* OR macroalg* OR “microalga*” OR “cyanobac*”) AND (“Marine”) AND (“bioactivit*” OR “Antioxidant*” OR “oxidati*” OR “inflammat*” OR “anti-inflammat*”) NOT (review). As filters, in Web of Science, were selected “Web of Science Core Collection—Open Access” and “Enriched References NOT Open Access”, whilst in Wiley Online Library the selected filter was “Journals”. In both databases, the filter “Last 5 years” was applied. References were organized in EndNote^TM^ version 21, and duplicates were removed using the same program. The initial screening was performed based on information available in the titles and abstracts of the papers.

All available studies that assessed the anti-inflammatory and antioxidant effect of marine algae on human health, both in vitro and in vivo studies, were included in our review. The inclusion criteria were articles that were published in the previous 5 years, in English, and assessed the anti-inflammatory and antioxidant effects of macroalgal, microalgal, or cyanobacterial extracts (or isolated compounds extracted from the abovementioned biomass sources) in diseases related to human health. The exclusion criteria were review articles, conference abstracts, studies performed with freshwater species, studies that had exclusively in chemico results or in vitro studies that did not display results as half-maximal inhibitory concentration (IC_50_ values), and studies on bioactivities not targeting human diseases (e.g., focusing on animal and plant health). The present review also eliminated research papers that utilized biomass sources other than algal biomass or involved interactions with compounds or products not of algal origin.

A total of 2927 studies were screened in the initial electronic search; 65 duplicated studies were found as well as 22 conference papers, which were both disregarded. After screening the titles and abstracts, 538 studies were considered suitable for retrieval, due to fulfilling the inclusion criteria. However, 77 papers could not be retrieved by EndNote (*n* = 461). After reviewing full-text articles, 296 were excluded for the following reasons: studies regarding bioactivities not relevant to the scope of this review (animal/plant, immunomodulatory, neurological, bone-related, or eye health) (*n* = 166), studies on food formulation and/or packaging including algae biomass, which did not contain relevant information for this review (*n* = 14), studies on biomasses other than marine algae or marine algae-derived compounds (*n* = 68), and studies presenting only in chemico results (*n* = 48). After this process, 165 studies were considered eligible for the review. Of the studies included in this review, a total of 72 had cell-based results and 32 studies were included by presenting in vitro results. Additionally, 61 in vivo studies showed the effects of marine algae compounds with several health-related bioactivities on humans, kittens, dogs, rats, mice, and zebrafish. When studies included more than one bioactivity, the study was only included in the section where the results were more significant. If several treatment models were tested, only the results from the higher model were included in the review, to avoid repetition. A flow diagram of the article selection process is shown in Figure 1. 

## 3. Health-Related Bioactivities

### 3.1. Antioxidant Properties

#### 3.1.1. Antioxidants and Their Role in Oxidative Stress and Disease Development

Free radicals, also known as oxidizing agents, are essential but unstable metabolic by-products of various normal cellular processes (e.g., mitochondrial respiration, inflammatory responses, and cellular signaling); however, their production can be increased due to exogenous stimuli (e.g., radiation, chemical agents, lifestyle factors, infections, and inflammation). The main oxidizing species are reactive oxygen species (ROS) such as hydroxyl radical (•OH), super oxide anion (O_2_^−^), hydrogen peroxide (H_2_O_2_), and singlet oxygen (O_2_) and reactive nitrogen species (RNS) like nitroxyl anion (NO^−^), nitrosonium cation (NO^+^), and various nitric oxide (•NO)-derived compounds, which are produced by the inducible nitric oxide synthase (iNOS) [29,30].

Antioxidants play a crucial role in preventing and repairing damages caused by ROS and RNS, by either stabilizing or eliminating them and preventing the oxidation of other molecules. Under normal circumstances, cellular endogenous antioxidant systems, which consist of enzymes (e.g., superoxide dismutase (SOD), catalase (CAT), and glutathione peroxidase (GPx)), and non-enzymatic biomolecules (such as bilirubin, ascorbate, glutathione, and albumin), are enough to safeguard against these reactive species and maintain a proper balance between oxidants and antioxidants. However, if there is an excessive production of ROS, the ability of cells to produce sufficient antioxidants may be compromised, resulting in inadequate protection for the organism. Under such conditions, the body can also resort to exogenous antioxidants sources such as food or nutraceuticals in an effort to restore equilibrium [30]. Nevertheless, if there is a persistent disturbance in the equilibrium between oxidants and antioxidants, it can result in the development of oxidative stress. Long-term oxidative stress can cause a chain reaction that plays a crucial role in cell damage and potential cell death (apoptosis), affecting various components like membranes, lipids, nucleic acids, and proteins [31]. Biomarkers for oxidative stress-induced damage include mutagenic and cytotoxic degradation products like malondialdehyde (MDA), resulting from lipid peroxidation, and deoxyguanosine (8-OHdG), resulting from DNA oxidation [29]. Oxidative stress can depolarize mitochondrial membranes and promote the release of pro-apoptotic proteins like cytochrome *c* into the cytosol. Cytochrome *c* associates with pro-caspase-9 and apoptosis activating factor-1 in the cytosol, activating caspase-9, which activates effector caspases (caspase-3, -6, and -7) to cleave cellular proteins and cause apoptosis. Other mitochondrial ROS-mediated apoptosis pathways involve DNA fragmentation, chromatin condensation, and the activation of the p53 and/or Jun N-terminal kinase (JNK) pathways, promoting intrinsic apoptosis (e.g., by activating pro-apoptotic Bax proteins and inhibiting anti-apoptotic proteins like Bcl-2) [32]. ROS activates apoptosis-related signaling pathways and transcription factors, including phosphoinositide 3-kinase (PI3K)/protein kinase B (Akt), mitogen-activated protein kinase (MAPK), nuclear factor erythroid 2–related factor 2 (Nrf2)/Kelch-like-ECH-associated protein 1 (Keap1), and nuclear factor kappa-B (NF-κB). When the Nrf2/Heme oxygenase-1 (HO-1) pathway is activated, Nrf2’s phosphorylation (mainly through kinases like MAPKs and PI3K) dissociates it from the Nrf2-Keap1 complex, allowing it to translocate to the nucleus and promote the expression of antioxidant response element (ARE) genes, controlling the expression of several enzymatic antioxidants like heme oxygenase 1 and increasing the cellular defense against oxidative stress [33]. If not properly regulated, oxidative stress can not only lead to the development of acute pathologies and premature aging, but also to chronic and degenerative conditions such as cancer and inflammatory, cardiovascular, gastrointestinal, and metabolic disorders [29]. Given the link between oxidative stress and these non-communicable diseases, antioxidant therapy or supplementation shows great potential in maintaining a healthy redox balance, postponing aging, and preventing or mitigating various health-related issues [30].

#### 3.1.2. Potential Health Benefits of Algal Antioxidants

A total of 13 cell experiments (Table 1) and 11 in vivo experiments (Table 2) fulfilled the inclusion criteria in this review regarding antioxidant properties of algae. All of them concern macroalgae or pure compounds of not-detailed algal origin. Many algal compounds exert their antioxidant effect by increasing the activity of endogenous antioxidant enzymes like SOD, CAT, and GPx [34], decreasing ROS production [35,36], and/or decreasing the expression of iNOS, leading to a decreased production of NO [37]. In skeletal myoblasts, algal compounds (phloroglucinol and Indole-6-carboxaldehyde) exerted antioxidant effects by regulating oxidative stress-induced apoptosis, mainly by decreasing mitochondrial dysfunction and modulating apoptosis-regulatory factors (e.g., caspases, Bcl-2, Bax, and cytochrome *c*) [38,39]. Some algal compounds (e.g., from *Ecklonia cava* and *Sargassum thunbergii*) were also able to decrease oxidative stress responses by modulating the signaling pathways, mainly by downregulating AMP-activated protein kinase (AMPK) and NF-κB, and upregulating Nrf2/ARE/HO-1/AKT pathways [38,39,40]. 

When studying skin conditions, matrix metalloproteinases (MMPs) emerged as pivotal players, due to their role in the degradation of the different components of the extracellular matrix. In the presence of oxidative stress (e.g., due to radiation exposure), the control of MMPs is disturbed, causing the breakdown of collagen and the extracellular matrix, resulting in visible skin aging signs like wrinkles and loss of elasticity, besides also contributing to the development of various skin disorders [41]. The use of algal compounds (6,6′-bieckol; bromophenol bis (2,3,6-tribromo-4,5-dihydroxybenzyl) ether; 3-Bromo-4,5-dihydroxybenzaldehyde; bromophenols; and phloroglucinol) seems to be a successful strategy to counteract oxidative stress-induced skin damage [42,43,44,45] and enhance overall skin health by maintaining the integrity of the matrix and decreasing the deregulation of MMPs [46]. The skin protective effect from extracts from *Fucus spiralis* [47] (in HaCaT cells) can potentially be attributed to its phlorotannin content, whilst the *Sargassum thunbergii* extract had 14 phenolic compounds in its composition [48].

**Table 1 molecules-29-01900-t001:** Cell experiments regarding algal extracts/compounds with antioxidant activities.

Complication	Algae Type	Algae Species	Algal Extract or Compound	Cell Line	Oxidative Stress Induced by	Concentrations Tested	Outcomes and Mechanism	References
n.d.	Macroalgae	*Ulva pertusa*	Ulvan	RAW 264.7	H_2_O_2_	200 µg/mL	↑ antioxidant activity (↑ CAT and SOD); ↑ expression of antioxidant genes (↑ GST, CAT, MnSOD, and GPx mRNA expression)	[34]
		*Undaria pinnatifida*	Phlorotannin extract	RAW 264.7	H_2_O_2_	10, 20, and 40 µg/mL	↑ cell survival; ↓ NO production and iNOS protein expression	[37]
Liver	Macroalgae	*Nizamuddinia zanardinii*	Fucoidan	HepG2	H_2_O_2_	0.1, 0.2, 0.5, and 0.7 µg/mL	Protective effect on H_2_O_2_-induced cytotoxicity; ↓ intracellular H_2_O_2_-induced ROS production; ↓ H_2_O_2_-induced damages	[49]
		*Pyropia haitanensis*	Floridoside	L-02	n.d.	200 µmol/L	No cytotoxic effect; ↑ SOD and GSH-Px activity; activation of HO-1 expression via upregulation on Nrf2/ARE and p38/ERK MAPK-Nrf2 pathway	[40]
Lungs	Macroalgae	*Gelidiella acerosa*	Ethyl acetate extract	A549	H_2_O_2_	1.5 mg/mL	↑ SOD and peroxidase activity	[50]
Skeletal muscle	Macroalgae	*Ecklonia cava*	Phloroglucinol	C2C12	H_2_O_2_	10 and 20 µg/mL	↓ cell toxicity (↓ H_2_O_2_-induced cell death); ↓ apoptosis (↓ DNA fragmentation, nuclear fragmentation, and chromatin condensation); ↓ mitochondrial dysfunction; regulation of apoptosis regulatory factors (↑ cytochrome *c* in the mitochondria, ↑ Bcl-2 expression, and ↑ caspase-3); ↓ ROS H_2_O_2_-induced accumulation; upregulation of Nrf2/HO-1 signaling pathway	[38]
	Macroalgae	*Sargassum thunbergii*	Indole-6-carboxaldehyde	C2C12	H_2_O_2_	400 µM	↓ cell toxicity (↓ H_2_O_2_-induced cell death); ↓ ROS overproduction; ↓ DNA damage; ↓ apoptosis; ↓ mitochondrial dysfunction; regulation of apoptosis regulatory factors (cytochrome *c*, Bax, Bcl-2, and caspase-3 and -9); downregulation of AMPK signaling pathway	[39]
Skin	Macroalgae	*Ecklonia cava*	6,6′-bieckol	HaCaT	UVB radiation	50 and 100 µM	↑ cell survival; antioxidant effect (↑ antioxidant enzymes); downregulation of matrix metalloproteinases (MMPs) through MAPK and NF-κB pathways	[46]
		*Fucus spiralis*	Ethyl acetate, water, and ethanol extracts	HaCaT	UVB radiation or H_2_O_2_	1000 µg/mL	↓ ROS production	[47]
		*Symphyocladia latiuscula*	Bromophenol bis (2,3,6-tribromo-4,5-dihydroxybenzyl) ether (BTDE)	HaCaT; HUVEC	H_2_O_2_	5 and 10 µM	↑ cell survival (↓ apoptosis); reverse oxidative damage induced by H_2_O_2_ (↓ ROS generation, ↓ MDA level, ↓ GSSG/GSH, and ↑ SOD activity); upregulation of Nrf2 and decrease in Keap1 expression; activation of AKT signaling pathway	[42]
	n.d.	n.d.	3-Bromo-4,5-dihydroxybenzaldehyde	HaCaT	H_2_O_2_ or UV-B radiation	30 µM	Protective effect against oxidative stress (↑ cell viability) possibly regulated by ERK and Akt pathways, inducing HO-1 and Nrf2 expression	[43]
			Bromophenols	HaCaT	H_2_O_2_	10 µM	↑ cell survival (↓ apoptosis); ↓ oxidative cell damage (↓ ROS generation); increased expression of antioxidant proteins (TrxR1 and HO-1)	[44]
			Phloroglucinol	HaCaT	H_2_O_2_	50 µM	Protected cells from H_2_O_2_-induced cytotoxicity (↑ cell viability); upregulation of Nrf2/HO-1 signaling pathway; ↓ oxidative stress (↓ ROS generation and DNA damage); ↓ apoptosis (↓ mitochondrial dysfunction); modulation of apoptosis regulatory genes (↑ Bcl-2, ↑ PARP, ↓ Bax, and ↑ caspase-3 and -9 expression); ↓ release of mitochondrial cytochrome *c* into the cytoplasm	[45]

AMPK: AMP-activated protein kinase; ARE: Antioxidant response element; Bax: Bcl-2-associated X protein; Bcl-2: B-cell lymphoma 2; DNA: Deoxyribonucleic acid; GPx: Glutathione peroxidase; GSH: Glutathione (reduced); GSSG: Glutathione disulfide; GST: Glutathione S-transferase; H_2_O_2_: Hydrogen peroxide; HO-1: Heme oxygenase-1; iNOS: Inducible nitric oxide synthase; MDA: Malondialdehyde; MMPs: Matrix metalloproteinases; n.d.: No data available; NF-κB: Nuclear factor kappa B; Nrf2: Nuclear factor (erythroid-derived 2)-like 2; NO: Nitric oxide; PARP: Poly(ADP-ribose) polymerase; ROS: Reactive oxygen species; SOD: Superoxide dismutase; ↑: Increase; ↓: Decrease.

Most in vivo experiments (Table 2) were conducted on zebrafish embryos, a well-established model system, including as a preclinical screening model. Results showed that extracts and isolated compounds from *Fucus virsoides* [35], *Gracilaria lemaneiformis* [36] (oligosaccharide), *Hizikia fusiforme* [51] (polysaccharide), *Padina boryana* [52], *Pyropia yezoensis* [53] (polyphenol), *Sargassum fulvellum* [54] (polysaccharide), and *Undaria pinnatifida* sporophylls [55] (polysaccharide) were successful in decreasing oxidative stress by reducing lipid peroxidation and ROS production, leading to decreased heart-beating disorders and increased survival rates. The liver and kidneys are particularly susceptible to the deleterious effects of ROS/RNS due to their elevated metabolic and mitochondrial activity. Hence, the monitoring of hepatic enzymes (e.g., aspartate aminotransferase, alanine aminotransferase, and alkaline phosphatase) and kidney function markers (e.g., urea and creatinine) in the serum is a common approach to assess potential liver injury and kidney impairment [56,57]. The methanol extract from *Halamphora* sp. exhibited a notable ability to reduce the concentration of liver enzymes in the bloodstream, indicating a hepatoprotective effect against oxidative stress-induced injuries [58]. This was further confirmed by improved hepatocyte histology, which might be associated with the high fatty acid (mainly palmitic and palmitoleic acid) content of the extract. Improved renal function markers were observed, as well as improved renal histology, suggesting a renal protective effect from the abovementioned extract and polysaccharide extract of *Ulva lactuca* [59]. Phenolic compounds extracted from *Sargassum thunbergii* [48], such as benzene and its derivatives (protocatechuic acid, difucol, gallic acid, and 4-hydroxybenzoic acid), cinnamic acids and their derivatives (p-Coumaric acid), flavonoids (isoquercitrin, quercitrin, isorhamnetin, and catechin), and phlorotannins (bifuhalol, pentafuhalol A, 7-hydroxyeckol, deshydroxypentafuhalol, trifuhalol A), along with a sulfated polysaccharide from *Ecklonia maxima* [60], were found to reduce the production of reactive oxygen species (ROS) and repair skin damage. Therefore, the identified algal compounds and extracts seem to have antioxidant properties [35,36,51,52,53,54,55], being able not only to restore a healthy balance between oxidants and antioxidants, but also aid in the regulation and mitigation of oxidative stress-induced damages in specific organs, such as the skin [48,60], liver, and kidney [58,59]. 

**Table 2 molecules-29-01900-t002:** In vivo experiments regarding algal extracts/compounds with antioxidant activities.

Complication	Algae Type	Algae Species	Algal Extraction or Compound	Route of Administration	Dosage	Experimental Period	Animal Model (Age)	Oxidative Stress Induced by	*n*/Group	Outcomes and Mechanism	References
n.d.	Macroalgae	*Fucus virsoides*	Less polar fractions	Incubation with embryo media	7.5, 15, and 30 µg/mL	4 d	Zebrafish embryos	H_2_O_2_	30	Decreased heartbeat frequency; ↓ ROS formation	[35]
		*Gracilaria lemaneiformis*	Agaro-oligosaccharides prepared from the agar	Incubation with embryo media	25 and 50 µg/mL	3 d	Zebrafish embryos	H_2_O_2_	n.d.	Increased survival rate (↓ cell death); ↓ heart-beating disorder; ↓ ROS production; ↓ lipid peroxidation	[36]
		*Hizikia fusiforme*	Fucoidan	Incubation with embryo media	25 and 50 µg/mL	2 d	Zebrafish embryos	H_2_O_2_	15	Increased survival rate (↓ cell death); ↓ heart-beating disorder; ↓ ROS production; ↓ lipid peroxidation	[51]
		*Padina boryana*	Ethanol precipitation	Incubation with embryo media	50 and 100 µg/mL	3 d	Zebrafish embryos (7–9 hpf)	H_2_O_2_	n.d.	Increased survival rate (↓ cell death); improved heart-beating rates; ↓ intracellular ROS; ↓ lipid peroxidation	[52]
		*Pyropia yezoensis*	Polyphenols and protein-rich extracts	Incubation with embryo media	12.5, 25, and 50 µg/mL	1 d	Zebrafish embryos (7–9 hpf)	AAPH	15	Decreased cell death; ↓ ROS production; ↓ lipid peroxidation production	[53]
		*Sargassum fulvellum*	Polysaccharides	Incubation with embryo media	50 and 100 µg/mL	3 d	Zebrafish embryos (7–9 hpf)	AAPH	15	Increased survival rate (↓ cell death); improved heart rate; ↓ intracellular ROS; ↓ lipid peroxidation	[54]
		*Undaria pinnatifida sporophylls*	Fucoidan	Incubation with embryo media	125 and 250 µg/mL	7 d	Zebrafish embryos (8 hpf)	AAPH	15	Increased survival rate (↓ cell death); ↓ heartbeat rate; ↓ ROS production; ↓ lipid peroxidation	[55]
Kidney	Macroalgae	*Ulva lactuca*	Polysaccharide extract	Intragastric	50 and 300 mg/kg	10 w	Kunming mice (8 w)	D-gal and ascorbic acid (subcutaneously)	9	Protective effect on kidney injury (↓ atrophy, ↓ serum creatinine and serum cystatin C); ↓ oxidative stress in kidney (↓ MDA, protein carbonyl, and 8-OHdG levels, and ↑ SOD, GSH-Px, and T-AOC); ↓ apoptosis (↓ expression of caspase-3 in kidney)	[59]
Liver and Kidney	Macroalgae	*Halamphora* sp.	Methanol extract (80%)	Gastric gavage	2 mg/kg/day	3 w	Wistar albino rats (adults)	Lead acetate (i.p.)	6	↓ lipid peroxidation in liver and kidney (↓ MDA); ↑ protection against oxidative stress in liver and kidneys (↑ GPx, SOD, and CAT); improved serum biochemical parameters (↓ AST, ALT, ALP, and LDH, and ↓ creatine and urea)	[58]
Skin	Macroalgae	*Ecklonia maxima*	Sulfated polysaccharides	Incubation with embryo media	50 and 100 µg/mL	3 d	Zebrafish embryos (7–9 hpf)	AAPH	15	↑ survival rate (↓ cell death, ↓ apoptosis); improved heart beating disorder; ↓ oxidative stress (↓ ROS generation and ↓ lipid peroxidation)	[60]
								UVB-exposure	10	↓ intracellular ROS levels; ↓ cell death; ↓ NO production and lipid peroxidation; improved collagen content and inhibition of MMPs	
		*Sargassum thunbergii*	Phenolic-rich extract	Incubation with embryo media	1.67 µg/mL	6 d	Zebrafish embryos (2 dpf)	UVB-exposure	8 to 10	Repaired skin damage; ↓ intracellular ROS accumulation	[48]

8-OHdG: 8-hydroxylated deoxyguanosine; AAPH: 2,2′-azobis (2-amidinopropane) dihydrochloride; ALT: Alanine aminotransferase; ALP: Alkaline phosphatase; AST: Aspartate aminotransferase; GPx: Glutathione peroxidase; H_2_O_2_: Hydrogen peroxide; hpf: Hours post fertilization; LDH: Lactate dehydrogenase; MDA: Malondialdehyde; MMPs: Matrix metalloproteinase; n.d.: No data available; NO: Nitric oxide; ROS: Reactive oxygen species; SOD: Superoxide dismutase; T-AOC: Total antioxidant capacity; ↑: Increase; ↓: Decrease.

### 3.2. Anti-Inflammatory Effects

#### 3.2.1. Inflammation and Its Role in the Onset and Progression of Diseases

Inflammation is a fundamental and intricate biological process that plays a vital role in maintaining the body’s homeostasis. An acute inflammatory response is triggered by either tissue injury or exposure to external stimuli (e.g., viruses or allergens). This response is initiated by various mediators, including cytokines like interleukins (IL) or tumor necrosis factors (TNFs), acute phase proteins (e.g., C-reactive protein), chemokines (e.g., Monocyte Chemoattractant Protein-1), or prostaglandins (PGE). These mediators facilitate the movement of immune cells (neutrophils and macrophages) to the site of inflammation by promoting vasodilation and angiogenesis, which allow for the migration of additional inflammatory cells. Usually, once the episode is resolved, inflammation is no longer needed. However, in some cases, inflammation can persist at low levels without any apparent cause, leading to chronic and uncontrolled inflammatory conditions, which has been linked to the development of various human diseases and disorders [61]. A complex interplay between oxidative stress and inflammation has been established, where the activation of the inflammatory cascade leads to the production of inflammatory mediators, causing oxidative stress, which in turn activates the inflammatory cascades [62]. 

#### 3.2.2. Mechanisms of Inflammation Modulation

The three most important intracellular inflammatory signaling pathways include the mitogen-activated protein kinase (MAPK), nuclear factor kappa-B (NF-κB), and Janus kinase (JAK)-signal transducer and activator of transcription (STAT) pathways. These pathways regulate pro-inflammatory cytokine production and inflammatory cell recruitment, which contribute to the inflammatory response [63]. The activation of the MAPKs, including Erk1/2, p38 and JNK, leads to the phosphorylation and activation of transcription factors, regulating pro-inflammatory gene expression, which initiates the inflammatory response (e.g., the expression of cytokines, chemokines, and inflammatory mediators). The activation of MAPK pathway is also linked to NF-κB and phosphoinositide 3-kinase (PI3K) pathways, as the MAPK mediates the phosphorylation of IκB kinase (IKK), which undergoes proteasomal degradation. This allows the NF-κB heterodimer (p50/p65) to translocate into the nucleus, bind to DNA, and induce target pro-inflammatory transcription. Meanwhile, the JAK/STAT signaling pathway is mostly activated by ligands (e.g., interleukins), activating the direct translation of an extracellular signal into a transcriptional response and controlling inflammatory gene transcription [63,64]. Several of these pathways are concurrently triggered by inflammatory mediators, regulating the expression of pro-inflammatory genes and ultimately leading to the synthesis of inflammatory mediators. In chronic inflammation, this becomes a positive feedback loop, leading to pathophysiological events [61,65]. 

Additional molecules that can modulate the inflammatory response are the arachidonic acid cascade-related eicosanoids (prostaglandins, thromboxanes, and leukotrienes). After phospholipases release arachidonic acid from the plasma membrane, cyclooxygenase (COX) or lipoxygenase (LOX) enzymes metabolize it, producing bioactive lipid mediators which act as signaling molecules. While COX-1 and some LOX are involved in normal cellular homeostasis, COX-2 is an inducible enzyme and, together with LOX-5, is upregulated in response to inflammatory stimuli, leading to the overexpressing of pro-inflammatory mediators, further increasing the inflammatory event; it is also overexpressed in pathophysiological events. Therefore, COX and LOX might be attractive therapeutic targets [66]. Nonsteroidal anti-inflammatory drugs (NSAIDs) are widely prescribed medications that reduce inflammation by blocking the COX enzyme. However, NSAIDs like ibuprofen and celecoxib can also have significant adverse side effects on the gastrointestinal, cardiovascular, hepatic, renal, cerebral, and pulmonary systems [21,67]. 

#### 3.2.3. Algal Applications in Managing Inflammatory Conditions

The anti-inflammatory activity of algae was found in a total of 48 studies—6 in vitro (Table 3), 31 cell experiments (Table 4), and 10 in vivo (Table 5). 

The in vitro anti-inflammatory studies that fitted the inclusion criteria for this review were all of macroalgal origin, with most of them focusing on extracts. Purified compounds were only obtained from two species, *Gracilaria salicornia* and *Turbinaria decurrens*, the former containing two 2H-chromenyl derivatives [68], two spiro-compounds [69], and a abeo-labdane type diterpenoid [70], whereas the latter accumulated a triterpene compound [71]). In terms of extracts, the anti-inflammatory activity (measured by the inhibition of inflammatory-inducing enzymes—COX and LOX) was most pronounced in *Gloeothece* sp. [72], *Gracilaria salicornia*, and *Padina tetrastromatica* Hauck [73], as these presented the lowest IC_50_ values compared to the other species. Notably, the anti-inflammatory effects observed in *Gloeothece* sp. were 10–20 times lower than those observed in the remaining species, showing that values not only vary significantly among species [73], but also according to extraction solvents [72]. 

**Table 3 molecules-29-01900-t003:** In vitro studies regarding algal extracts/compounds with anti-inflammatory activities.

Algae Type	Algae Strain	Type of Analyzed Sample (Extract or Pure Compound)	In Vitro Assays Against Pro-Inflammatory Enzymes (IC_50_ Values in µg/mL Unless Otherwise Stated)			References
			COX-1	COX-2	5-LOX	
Macroalgae	*Amphiroa fragilíssima* (Linnaeus) J.V. Lamouroux	EtOAc-MeOH extracts	4990	5010	5020	[73]
	*Gloeothece* sp.	Acetone Ethanol Hexane:isopropanol (3:2)		120 200 130		[72]
	*Gracilaria canaliculata* Sonder	EtOAc-MeOH extracts	2920	2000	2010	[73]
	*Gracilaria corticata* (J. Agardh) J. Agardh		2990	3010	3020	
	*Gracilaria salicornia*	4′-[10′-[7-hydroxy-2,8-dimethyl-6-(pentyloxy)- 2*H*-chromen-2-yl]ethyl]-3′,4′-dimethyl-cyclohexanone			2.46 mM	[68]
		3′-[10′-(8-hydroxy-5-methoxy-2,6,7-trimethyl-2*H*-chromen2-yl)ethyl]-3′-methyl-2′-methylene cyclohexyl butyrate			2.03 mM	
	*Gracilaria salicornia*	spiro[5.5]undecanes, 3-(hydroxymethyl)-7-(methoxymethyl)-3,11-dimethyl-9-oxospiro[5.5]undec-4-en-10-methylbutanoate			2.78 mM	[69]
		4-ethoxy-11,11-dimethyl-7-methylene-8-(propionyloxy)spiro[5.5]undec-2-en-10^4^,10^6^-dihydroxytetrahydro-2*H*-pyran-10-carboxylate			1.91 mM	
	*Gracilaria salicornia*	Methyl-16(13→14)-abeo-7-labdene-(12-oxo) carboxylate			860	[70]
	*Gracilaria salicornia*	EtOAc-MeOH extracts	1010	1020	980	[73]
	*Halymenia dilatata* Zanardini		3040	3000	3020	
	*Hydropuntia edulis* (S.G.Gmelin) Gurgel & Fredericq		2910	3010	2980	
	*Padina tetrastromatica* Hauck		1230	1340	1280	
	*Palisada pedrochei* J.N.Norris		4040	4030	4010	
	*Portieria hornemannii* (Lyngbye) P.C. Silva		2010	1990	2030	
	*Spyridia filamentosa* (Wulfen) Harvey		3010	2980	3040	
	*Turbinaria decurrens*	Decurrencyclic B		14.0 µM	3.0 µM	[71]

EtOAc: Ethyl acetate; COX: Cyclooxygenase; LOX: Lipoxygenase; MeOH: Methanol.

Most cell studies evaluated the effect of algal compounds and extracts on general inflammation in macrophage cell models (e.g., RAW 264.7) (*n* = 20), with compounds being predominantly of macroalgal origin (*n* = 18) and only two derived from microalgae (*Phaeodactylum tricornutum* [74] and *Tisochrysis lutea* [75]). The algal compounds acted through several mechanisms: decreased oxidative stress (by decreasing ROS and NO production or upregulating the Nrf2/HO-1 pathway); a decreased activity of inflammation enzymes (COX, iNOS); decrease in inflammatory transcription factor (NF-κB) levels; a decreased expression and production of pro-inflammatory chemokines and cytokines (such as interleukin (IL)-6, IL-1beta (IL-1β), TNF-alpha (TNF-α), and prostaglandins (PGE)); an increased expression and production of anti-inflammatory cytokines (IL-4 and IL-10); and a decreased upregulation of inflammatory pathways such as MAPK, NF-κB, and JAK/STAT. These findings were transversal regardless of treating general inflammation or specific disorders such as skin diseases (e.g., atopic dermatitis) and inflammatory myopathy. Interestingly, the ethanol extract from a combination of *Ecklonia cava* and *Sargassum horneri* [76] exhibited a more effective anti-inflammatory effect used in combination than individual macroalgae extracts, which might be attributed to synergistic effects. Several aspects influence the algal extract’s activity; for example, when evaluating lipid crude extracts from *Sargassum ilicifolium,* it was concluded that the anti-inflammatory activity is higher in a preventive scenario rather than in a treatment approach [77], whilst when evaluating the ethyl acetate fraction of *Himanthalia elongate* [78], it was discovered that the anti-inflammatory activity in the digested sample was increased in comparison to crude extracts, possibly due to the breakdown of complex phlorotannin structures.

Cell assays also focused on skin-related disorders, following consumers’ rising aware of skin aging and search for ways to counteract this trend with novel active ingredients [79]. Besides exerting anti-inflammatory effects through the abovementioned general anti-inflammatory mechanisms, some macroalgal compounds additionally had a protective activity of the skin barrier [80], decreased wrinkle formation [81], increased cell proliferation and collagen production in human dermal fibroblasts [82], and downregulated the expression of MMPs [83], contributing to overall skin health.

**Table 4 molecules-29-01900-t004:** Cell studies regarding algal extracts/compounds with anti-inflammatory activities.

Complication	Algae Type	Algae Species	Algal Extraction or Compound	Cell Line	Inflammation Induced by	Concentrations	Outcomes and Mechanism	References
n.d.	Macroalgae	*Caulerpa racemosa*	Ethanol, hexane, and ethyl acetate carotenoid fractions	RAW 264.7	LPS	25 µM	↑ AMPK expression; ↓ TNF-α expression; ↓ mTOR expression	[84]
		*Cystoseira amentacea*	Ethanol or DMSO extract	RAW 264.7	LPS	100 µg/mL	↓ inflammation (↓ IL-1β, IL-6, COX-2, and iNOS expression)	[85]
		*Dictyopteris membranacea*	Disulfides	RAW 264.7	LPS	15.62–31.25 µM	Anti-inflammatory activity (↓ TNF-α, IL-6, and IL-12 production); ↓ NO expression by downregulating iNOS; downregulation of AKT/MAPK/ERK signaling pathway	[86]
		*Ecklonia cava* and *Sargassum horneri*	Ethanol extract	RAW 264.7	LPS	62.5 µg/mL	No cytotoxic effect; ↓ NO production; ↓ inflammatory response (↓ IL-1β, IL-6, PGE2, and TNF-α expression); downregulation of iNOS and COX-2; downregulation of NF-κB and MAPK pathways	[76]
		*Ecklonia cava*	Ethanol extract	HGF-1	LPS	50 and 100 µg/mL	↓ PGE2 production and pro-inflammatory enzyme expression; ↓ pro-inflammatory chemokine gene expressions; ↓ ROS production; downregulation of MAPK signaling pathway	[87]
		*Himanthalia elongata*	Ethyl acetate fraction of a crude acetone extract	RAW 264.7	LPS	100 µg/mL	↓ NO and O_2_ production regardless of being submitted to a simulated gastrointestinal digestion or not	[78]
		*Laurencia majuscula*	Sesquiterpene (C_17_H_25_BrO_3_); chamigrane	RAW 264.7	LPS	3.7 µM; 3.6 µM	↓ NO production and no cytostatic activity	[88]
		*Padina boryana*	Fucosterol	RAW 264.7	Particulate Matter/LPS	12.5, 25, and 50 µg/mL	↓ NO production; ↓ cytokines production (↓ IL-1β, IL-6, TNF-α, and PGE2); ↓ mRNA expression of IL-1β, IL-6, TNF-α, iNOS, and COX-2; downregulation of MAPK and NF-κB phosphorylation; upregulation of Nrf2/HO-1 pathway	[89]
		*Porphyra tenera*	Water extract	RAW 264.7	LPS	1000 µg/mL	↓ PGE2 and NO production; ↓ COX-2 and iNOS protein expression; ↓ TNF-α and IL-6 production	[90]
		*Porphyra* sp.	Polydeoxyribonucleotide	RAW 264.7	LPS	200 µg/mL	↓ NO production; ↓ iNOS expression by reducing phosphorylation of p38 MAPK and ERK	[82]
		*Rugulopteryx okamurae*	Rugukadiol A and ruguloptone A	RAW 264.7	LPS	10 µM	↓ NO production; ↓ Nos2 and IL-1β expression	[91]
		*Rugulopteryx okamurae*	Okaspatol C Okamurol A	RAW 264.7	LPS	10 µM	Decrease in NO production	[92]
		*Sargassum autumnale*	Fucoidan fractions	RAW 264.7	LPS	50, 100, and 200 µg/mL	↓ NO production (↑ cell viability); ↓ PGE2 production; ↓ pro-inflammatory cytokines (TNF-α, IL-6, and IL-1β); ↓ expression of inducible inflammatory enzymes (iNOS and COX2); downregulation of NF-κB and MAPK pathways	[93]
		*Sargassum horneri*	Sargachromenol	RAW 264.7	LPS	62.5 µg/mL	↑ antioxidant activity (↓ NO and intracellular ROS production); activation of Nrf2/HO-1 signaling pathway (upregulation of HO-1 expression); ↓ expression of inflammatory cytokines (IL-1β, IL-6, and TNF-α) through the downregulation of iNOS and COX-2 expression; suppression of activation of NF-κB and MAPK signaling pathways	[94]
		*Sargassum ilicifolium*	Crude lipid extracts	RAW 264.7	LPS	50 µg/mL	↓ NO production in pre-incubated and co-incubated cell culture models	[77]
		*Saccharina japonica*	Fucoidan	RAW 264.7	LPS	100, 150, and 200 µg/mL	↓ NO production; ↓ inflammation (↓ iNOS and COX-2 expression and ↓ TNF-α, IL-6, and IL-1β production); downregulation of NF-κB, MAPK, and JAK2-STAT1/3 signaling pathways	[95]
		*Sargassum swartzii*	Fucoidan fraction	RAW 264.7	LPS	100 and 200 µg/mL	↓ NO production; ↓ inflammation (↓ PGE2, TNF-α, IL-1β, and IL-6 secretion and expression); ↓ iNOS and COX-2 expression; downregulation of NF-κB and MAPK signaling pathways	[96]
		n.d.	Fucoxanthinol	RAW 264.7	LPS	10 and 20 µM	Anti-inflammatory activity (↓ iNOS, IL-6, and TNF-α mRNA expression and ↓ IL-1β, TNF-α, IL-6, and Nitrate production)	[97]
	Microalgae	*Phaeodactylum tricornutum*	Nonyl8-acetoxy-6-methyloctanoate	RAW 264.7	LPS	25 μg/mL	↓ inflammation (↓ NO, PGE2, IL-1β, and IL-6); downregulation of COX-2 and iNOS	[74]
		*Tisochrysis lutea*	Methanol extract	RAW 264.7	LPS	100 µg/mL	Protected cells from cytotoxicity (↓ dendritic structures); ↓ PGE2 production and COX-2 protein expression; ↓ IL-6 and ↑ IL-10 expression; ↓ expression of inflammatory genes (Arg1, SOD2, and NLRP3)	[75]
Myopathy	Macroalgae	*Ishige okamurae*	Diphlorethohydroxycarmalol	C2C12	TNF-α	3.125, 6.25, and 12.5 µg/mL	↓ NO and ↓ pro-inflammatory cytokines (TNF-α, IL-1β, and IL-6) production; modulation of NF-κB and MAPK signaling pathways	[98]
Skin	Macroalgae	*Ecklonia cava*	Dieckol	HaCaT	Particulate matter	10 and 30 µM	↓ PGE2 production; ↓ COX-1 and COX-2 mRNA expression levels; ↓ ROS; ↓ gene expression of enzymes involved in PGE2 synthesis	[99]
		*Halymenia durvillei*	Ethyl acetate fraction	HaCaT	UV radiation	5 µg/mL	↓ intracellular ROS production; ↓ matrix metalloproteinases; upregulation of mRNA of antioxidant enzymes (SOD, HMOX1, and GSTP1); ↑ procollagen synthesis; activation of Nrf2 pathway	[81]
		*Polyopes affinis*	Butanol fraction	HaCaT	IFN-γ or TNF-α	10, 30, and 60 µg/mL	Downregulation of MAPK, STAT1, and NF-κB pathways	[100]
		*Polysiphonia morrowii*	3-bromo-4,5-dihydroxybenzaldehyde	HaCaT	IFN-γ or TNF-α	144 and 288 µM	↓ inflammatory cytokines (IL-6, IL-8, IL-13, IFN-y, and TNF-α) and chemokine production; downregulation of MAPK and NF-κB signaling pathways; activation of Nrf2/HO-1 signaling; protective activity against deterioration of skin barrier function (preserving skin moisture and tight junction stability)	[80]
		*Pyropia yezoensis*	Methanol extract	HaCaT	IFN-γ	40, 200, and 1000 µg/mL	Improvement of atopic dermatitis (↓ mRNA expression and secretion of pro-inflammatory chemokines; inhibition of MAPK activation; downregulation of NF-κB activation)	[101]
		*Sargassum confusum*	Low-molecular-weight fucoidan	HaCaT	IFN-γ or TNF-α	15.6, 31.3, and 62.5 µg/mL	↓ ROS production; ↓ inflammatory cytokines (IL-1β, IL-6, IL-8, IL-13, IFN-y, and TNF-α) and chemokines; downregulation of MAPK and NF-κB signaling pathways; activation of Nrf2/HO-1 signaling	[102]
		*Sargassum horneri*	(–)-Loliolide	HaCaT	IFN-γ or TNF-α	15.6, 31.3, and 62.5 µg/mL	↓ inflammatory cytokines (IL-4, IL-6, IL-13, IFN-y, and TNF-α) and chemokines; downregulation of MAPK and NF-κB signaling pathways; upregulation of Nrf2/HO-1 signaling	[103]
		*Sargassum siliquastrum*	Low-molecular-weight fucoidan	RAW 264.7	LPS	25, 50, and 100 µg/mL	↓ ROS production; ↓ production of NO and PGE2; ↓ expression of iNOS and COX-2; ↓ inflammatory cytokine expression (IL-1β, IL-6, and TNF-α); downregulation of MAPK and NF-κB signaling pathways; activation of Nrf2/HO-1 signaling; inhibition of the NLRP3 inflammasome protein complex	[104]
		*Sargassum horneri*	Fucosterol	HDF	IFN-γ or TNF-α	60 and 120 µM	↓ ROS production; activation of Nrf2/HO-1 signaling; no effect of cell viability; ↓ mRNA expressions of inflammatory cytokines (IL-6, IL-8, IL-13, IL-33, IL-1β, TNF-α, and IFN-y) and MMPs; downregulation of MAPK and NF-κB signaling pathways	[83]
	Microalgae	*Porphyridium cruentum*	Sulfated exopolysaccharides	HaCaT	UVA radiation	12 µg/mL	Protective effect on cells from oxidative damage (↓ ROS formation, ↓ lipid peroxidation, and ↑ intracellular GSH levels); increased wound healing activity	[105]
			Phycoerythrin			10 nM		

AKT: Protein kinase B; AMPK: AMP-activated protein kinase; COX: Cyclooxygenase; DNA: Deoxyribonucleic acid; DMSO: Dimethyl sulfoxide; ERK: Extracellular signal-regulated kinase; GSH: Glutathione; GST: Glutathione S-transferase; HGF-1: Human gingival fibroblast-1; HDF: Human dermal fibroblast; HO-1: Heme oxygenase-1; IFN-γ: Interferon-gamma; IL: Interleukin; iNOS: Inducible nitric oxide synthase; JAK2-STAT1/3: Janus kinase 2-signal transducer and activator of transcription 1/3; LPS: Lipopolysaccharide; MAPK: Mitogen-activated protein kinase; mTOR: Mammalian target of rapamycin; n.d.: No data available; NF-κB: Nuclear factor-kappa B; NLRP3: NOD-like receptor protein 3; NO: Nitric oxide; Nrf2: Nuclear factor erythroid 2–related factor 2; PGE2: Prostaglandin E2; ROS: Reactive oxygen species; SOD: Superoxide dismutase; STAT1: Signal transducer and activator of transcription 1; STAT3: Signal transducer and activator of transcription 3; TNF-α: Tumor necrosis factor-alpha; TXA2: Thromboxane A2; UVA: Ultraviolet A; UVB: Ultraviolet B; ↑: Increase; ↓: Decrease.

In vivo studies (Table 5) followed the same trend as cell experiments by mainly evaluating general inflammation (*n* = 7 for macroalgal and *n* = 2 for microalgal studies) and skin disorders (*n* = 2, 1 study for each algae type). The anti-inflammatory effects were attributed to the decrease in NO, ROS, and IL-1β, thus decreasing LPS-induced cell death. Interestingly, the two studies with human subjects were performed with microalgae, where *Phaeodactylum tricornutum* supplements [106] showed good tolerance and improved inflammatory status and the modulation of intestinal permeability, and the skin application of *Dunaliella salina* demonstrated anti-inflammatory and anti-aging effects [107]. Polysaccharides were the only class of isolated compounds that exhibited anti-inflammatory action [108,109,110,111,112].

**Table 5 molecules-29-01900-t005:** In vivo studies regarding algal extracts/compounds with anti-inflammatory activities.

Complication	Algae Type	Algae Species	Algal Extraction or Compound	Route of Administration	Dosage	Experimental Period	Animal Model (Age)	Inflammation Induced by	*n*/Group	Outcomes and Mechanism	References
n.d.	Macroalgae	*Codium fragile*	Sulfated polysaccharides	n.d.	50 and 100 µg/mL	3 d	Zebrafish embryos (7–9 hpf)	LPS (10 µg/mL)	n.d.	↓ cell death; ↓ NO and ROS generation	[108]
		*Cystoseira crinita (Desf.) Borry*	Fucoidan	Intraperitoneally	25 and 50 mg/kg	5 h	Wistar Rats	LPS (0.25 mg/kg)	8	Decrease in IL-1β production	[109]
		*Ecklonia maxima*	Ethyl acetate fraction	Incubation with embryo media	25 and 50 µg/mL	3 d	Zebrafish embryos (7–9 hpf)	LPS (10 µg/mL)	15	Increased survival rate (↓ cell death); improved heart-beating rates; ↓ ROS and NO generation	[113]
		*Saccharina japonica*	Sulfated polysaccharide	Incubation with embryo media	50 and 100 µg/mL	3 d	Zebrafish embryos (8 hpf)	LPS (10 µg/mL)	15	↓ cell death; ↓ NO and ROS generation; protection of phenotypic changes and toxic damages caused by LPS (↓ yolk sack edema, ↓ heart rate, and ↑ survival rate)	[114]
		*Saccharina japonica*	Fucoidan	Incubation with embryo media	25 and 50 μg/mL	3 d	Zebrafish embryos	LPS (10 µg/mL)	n.d.	Increased survival rate (↓ cell death); improved heart-beating rates; ↓ intracellular ROS; ↓ NO generation	[110]
		*Sargassum binderi*	Polysaccharides	Incubation with embryo media	25, 50, and 100 µg/mL	3 d	Zebrafish larvae (7–9 hpf)	LPS (10 µg/mL)	15	↓ LPS-induced cell death; ↓ NO production	[111]
		*Sargassum fulvellum*	Polysaccharides	Incubation with embryo media	50 and 100 µg/mL	3 d	Zebrafish embryos (7–9 hpf)	LPS (10 µg/mL)	15	Increased survival rate (↓ cell death); ↓ heartbeat disorder; ↓ ROS; ↓ NO	[112]
	Microalgae	*Phaeodactylum tricornutum*	Supplements (whole biomass, β-1,3-glucan-rich, or combination thereof)	Oral supplements	2.3 g biomass powder; 1.8 g of lyophilised supernatant; 2.3 g biomass powder + 1.8 g of lyophilised supernatant	2 w	Elderly human individuals (67.7 ± 6.5 years)	-	4 to 5	No severe reactions, some mild and minimal were reported; decreased inflammatory marker (IL-6); ↑ plasma carotenoids (fucoxanthin); modulation of intestinal permeability (↓ zonulin)	[106]
		*Tetraselmis* sp.	Ethanol extract	Incubation with embryo media	100 and 200 µg/mL	7 d	Zebrafish embryos (7–9 hpf)	LPS (10 µg/mL)	15	Increased survival rate (↓ cell death); ↓ NO generation	[115]
Skin	Macroalgae	*Sarcodia suiae* sp.	Ethyl acetate fraction of ethanol extract	Skin application	200 µg/day	18 d	BABL/c mice (8 w)	DNCB (2%)	6	↓ Atopic dermatitis symptoms (↓ inflammation, skin erythema, edema, dryness, and keratinocyte hyperplasia) and ↓ immunoglobulin E upregulation; ↓ swelling of subiliac lymph nodes and spleen; ↑ skin barrier integrity (↑ claudin-1 expression, cell-to-cell connections, and improved dilaggrin deficiency)	[116]
	Microalgae	*Dunaliella salina*	Hydrophobic extract	Skin application	1%	56 d	Human subjects (aged 35–60, Fitzpatrick skin phototypes II–IV, and with signs of aging)	Intense solar exposure	25	Anti-inflammatory activity (↓ skin reactivity to histamine stimulation and red spot count and area); anti-aging effect (↓ wrinkle count and volume)	[107]

IL: Interleukin; LPS: Lipopolysaccharide; n.d.: No data available; NO: Nitric oxide; ROS: Reactive oxygen species; w: Weeks; ↑: Increase; ↓: Decrease.

### 3.3. Cardioprotective Activity

#### 3.3.1. Cardiovascular Diseases and Regulation of Blood Pressure and Blood Lipid Levels

Cardiovascular diseases (CVDs) encompass a range of heart and blood vessel problems, which are the primary death cause worldwide [117,118]. The main risk factor for the development of CVDs is the inherent aging process of the cardiovascular system, where the main stress factors are oxidative stress and chronic inflammation, which interact in a positive feedback loop. ROS contributes to the development of myocardial tissue damage, modifies calcium homeostasis and contractile dysfunction, and causes cardiomyocyte hypertrophy, apoptosis, and fibrosis [119]. ROS also promotes the recruitment of inflammatory cells. It increases the expression of adhesion molecules like intercellular and vascular cell adhesion molecule 1 (ICAM-1 and VCAM-1, respectively), enabling lipids accumulation in the inner layer of blood vessels [120]. Therefore, the inflammatory cascade and ROS play an important role in the development, modulation, and progression of atherosclerotic plaque. Together with lipid core growth, a reduction in the thickness of the fibrous cap leads to plaque instability, significantly increasing the risk of rupture and potential acute events such as stroke [121]. Over time, these factors result in a gradual deterioration of physiological functions and to the development of disorders such as hypertension, heart failure, arteriosclerosis, atherosclerosis, and myocardial infarction [118,119].

Aside from the natural aging process (senescence), there are behavioral risk factors that increase the risk of developing CVDs, which may manifest clinically in individuals, such as hypertension, dyslipidemia, hyperglycemia, overweight, and obesity [117]. The angiotensin-converting enzyme (ACE) is a protease that regulates the renin–angiotensin–aldosterone system, which plays a vital role in maintaining circulatory hemodynamics and participating in cardiac aging. ACE converts inactive angiotensin I into angiotensin II (Ang II), a strong vasoconstrictor and salt retention promoter, which controls blood pressure. Therefore, the deregulation of ACE levels can cause hypertension and other cardiovascular issues such as cardiac hypertrophy and cardiomyocyte apoptosis. Furthermore, Ang II, via the AT1 receptor, contributes to heart inflammation by promoting IL-6, IL-1β, and TNF-α production. Hence, in certain cardiovascular disorders, it is advantageous to inhibit the renin–angiotensin–aldosterone system, thereby preventing blood vessel constriction, reducing blood pressure, and suppressing the production of inflammatory cytokines [119,122]. Dyslipidemia, a contributing factor to the development of cardiovascular diseases, primarily arises from elevated levels of total cholesterol (TC), triglycerides (TG), low-density lipoprotein cholesterol (LDL), and reduced levels of high-density lipoprotein (HDL). This condition negatively impacts blood vessel health by promoting the inflammation and oxidative stress damage of the endothelial cells that line the blood vessels, leading to atherosclerosis and the formation of plaque [118].

#### 3.3.2. Algal Compounds and Their Potential for Reducing Cardiovascular Risk

During the literature search, seven in vitro studies with cardioprotective activity fulfilled the inclusion criteria for this review. These studies evaluated the anti-hypertensive activity of macroalgal extracts, fractions, protein hydrolysates, peptides, and isolated compounds (chromenols from *Sargassum macrocarpum* [123]) by measuring the inhibition of the ACE (Table 6). ACE-inhibitory potential varied among species but also among extraction solvents [123]. When trying to further identify the bioactive compound of *Mazzaella japonica*, purification led to a loss of hypertensive activity, as the ACE inhibitory activity in each fraction ranged from 1.3 to 6.7% of the total ACE inhibition, suggesting that every fraction had a significant role in the overall ACE inhibitory activity [124]. In contrast, the hydrolysate from *Ulva intestinalis* exhibited the opposite behavior, where bioactivity was increased in the fraction with a molecular weight below 3 kDa [125], indicating a concentration of the bioactive compound(s) in this particular fraction.

Only two cell experiments reporting cardioprotective effects fulfilled the inclusion criteria for this review. A water extract from macroalgae *Fucus vesiculosus* (250 µg/mL) showed promising results for fighting hypercholesterolemia, by reducing cholesterol permeation in Caco-2 cells by almost 50% [129]. The other study focused on the potential of a nonapeptide (EMFGTSSET) extracted from microalgae *Isochrysis zhanjiangensis* [130], at 100 µM, to improve atherosclerosis in HUVEC cells stimulated by oxidized low-density lipoprotein. Results were promising by showing a reduction in oxidative stress and inflammatory markers (ROS, IL-6, IL-1β, and TNF-α), cell adhesion molecules (ICAM-1 and VCAM-1), apoptosis (less caspase-3 and -9 expression), downregulating inflammatory pathways (e.g., MAPK), and upregulating Nr2/HO-1 pathway.

Several in vivo studies (*n* = 6) were found in which macro- and microalgal (*n* = 5 and *n* = 1, respectively) compounds demonstrated the potential to reduce the risk of CVD through various mechanisms (Table 7). These include regulating blood lipid levels (increasing HLD and decreasing TC, LDL, and TG) [131], reducing plaque formation and/or instability [131], decreasing oxidative stress and inflammatory markers [132], modulating inflammatory pathways [132], improving blood pressure control [133,134], reducing cardiac fibrosis in the ventricle areas [135], ameliorating histological changes in the cardiac tissue [132], and mitigating senescent deterioration [135]. Hydrolysates [133,134], terpenes [132], a mix of a polysaccharide and a carotenoid [135], and isolated polysaccharides [131,135,136] showed favorable outcomes for hypertension [133,134], myocardial inflammation [132], aging [135], carotid atherosclerotic lesions [131], and heart valve calcification [136], respectively.

### 3.4. Gastrointestinal Health Modulation

#### 3.4.1. Implications for Digestive Health and Gut-Related Disorders

Gastrointestinal diseases are prevalent worldwide, resulting in a significant decline in patients’ quality of life, imposing a significant healthcare burden and expenses, and potentially resulting in mortality [137]. Gastrointestinal problems, such as inflammatory bowel diseases and liver disorders, not only have a direct influence on the affected organs but also have broader consequences, such as impairing gastrointestinal health and reducing the overall health and well-being of patients [138].

Inflammatory bowel disease (IBD) is a persistent inflammatory condition affecting the gastrointestinal system, caused by a multifaceted interplay of genetic, immunological, and environmental factors. Crohn’s disease and ulcerative colitis are the two main subtypes of IBD, differing not only in their location but also in the nature of inflammation. Ulcerative colitis predominantly impacts the colon, causing persistent inflammation and the formation of ulcers. In contrast, Crohn’s disease can affect any segment of the digestive system, characterized by sporadic but transmural bowel inflammation. Both disorders exhibit varying symptoms and risks, and presently, there is no cure available for either. However, a common hallmark of both conditions is dysbiosis in the gut microbiota [138]. In fact, alterations in the gut microbial ecology have been linked to several NCDs (e.g., colorectal cancer, atherosclerosis, diabetes, obesity, liver disorders, and osteoporosis), being postulated as the communicable element [139].

The gut microbiota is essential for maintaining a symbiotic relationship with the host, influencing multiple physiological processes such as digestion, nutrient absorption, and immune system regulation. A healthy host’s gut microbiota, mostly Firmicutes and Bacteroides, synthesizes vitamins that the host cannot produce (e.g., vitamin K) and ferments non-digestible dietary fibers to produce short-chain fatty acids (SCFAs), like acetate, propionate, and butyrate, which regulate immune response, inflammation, and gut barrier integrity. Additionally, the gut microbiota breaks down bile acids, which directly regulate intestinal immune cell populations and maintain mucosal barrier immunity. Gut dysbiosis disturbs the delicate balance of microbial ecology (diversity), which can result in inflammation, compromised intestinal barrier function, and modified metabolism, becoming a central factor in the development of several clinical disorders [138,140].

Gut dysbiosis leads to oxidative stress, which can abnormally activate the intestinal immune system and harm the intestinal mucosal barrier by reducing mucous secretion, antimicrobial peptide secretion, and impair tight junctions. The immune response will activate pro-inflammatory signaling pathways such as NF-κB, JAK/STAT, and MAPK, which result in the release of proinflammatory factors and oxidases (e.g., iNOS, COX-2, and NOX), leading to a positive feedback loop of oxidate stress, inflammation, and changes in the composition of the gut microbiota [141].

Liver disorders, such as non-alcoholic fatty liver disease (NAFLD), often coexist with IBD as comorbid conditions, likely due to the connection between the gut and the liver known as the “gut–liver axis” [138,139]. NAFLD is the most prevalent chronic liver disease worldwide and covers a wide spectrum of liver diseases ranging from simple steatosis to non-alcoholic steatohepatitis (NASH), liver fibrosis, and, ultimately, cirrhosis and hepatocellular carcinoma. Hepatic steatosis is caused by the excessive accumulation of fat in the liver, as well as lipogenesis and systemic insulin resistance. NASH develops gradually because of fat buildup, which disrupts metabolism and leads to an excessive production of ROS in the mitochondria. This, in turn, causes lipid peroxidation and reduces the levels of antioxidant enzymes in the liver. Oxidative stress also controls the activation of genes related to lipid metabolism, such as peroxisome proliferator-activated receptor (PPAR) and sterol regulatory element-binding protein (SREBP). The liver’s compromised metabolic function causes an excess buildup of free fatty acids (FFAs) within the liver cells, causing lipotoxicity, inflammation, cellular damage, and cell death. Hepatic damage can be reversed prior to the onset of fibrosis. The progression of these conditions will lead to chronic injury and the development of fibrosis, resulting in liver cirrhosis and, eventually, hepatocellular cancer [138,142].

#### 3.4.2. Algal Compounds and Their Potential for Improving the Impairment of Gastrointestinal Health

The literature search yielded only two cell assay experiments that fulfil the inclusion criteria of this review, both investigating the liver promoting activity of purified compounds (fucoidan and fucoxanthin) derived from macroalgae. The fucoidan (50 µg/mL) derived from *Cystoseira compressa* showed antisteatotic action by modulating the lipogenesis pathway (PPARy) and decreasing intracellular triglyceride content in FaO cells stimulated by oleate and palmitate [143]. The other macroalgae compound, fucoxanthin (25 and 50 µM), was able to mitigate zearalenone-induced hepatotoxicity by decreasing inflammation (less production of pro-inflammatory cytokines TNF-α, IL-6, and IL-1β) and oxidative stress (by reducing ROS production and upregulating the PI3K/AKT/NRF2 signaling pathways) in hepatocytes (HepG2 cells) [144].

A total of 14 in vivo studies were incorporated in this review to assess the impact of extracts, compounds, or commercial items on liver health (*n* = 4) or gut health (*n* = 10), all of macroalgal origin (Table 8). In terms of liver health, the treatment with algal compounds resulted in a reduction in body [145] and liver weight [146], indicating a decrease in fat accumulation and a decrease in liver steatosis [145]. Additionally, hepatic inflammation was reduced through the reduction in cytokine production and the modulation of inflammatory pathways [147]. The treatment also restored hepatic lipid metabolism by modulating genes involved in lipogenesis [147]. Furthermore, it decreased oxidative stress by increasing the profile of antioxidant liver enzymes and ultimately decreased liver damage, as evidenced by the decrease in liver enzymes in the serum and the improvement of the abnormal cellular architecture of liver tissue [146,148]. Hence, the algae-based compounds that were evaluated effectively influenced all the fundamental factors that contribute to the development and progression of NAFLD, suggesting their potential as a source for therapeutic interventions.

An *Ulva pertusa* extract also effectively alleviated the negative effects of IBD by providing the relief of symptoms such as pain, weight loss, and colon shortening [149,150]. Additionally, several compounds counteracted ulcerative colitis by decreasing the levels of pro-inflammatory cytokines and enzymes, while increasing the levels of anti-inflammatory cytokines and modulating inflammatory pathways [151,152,153,154,155]. Oxidative stress was reduced via enhancing the activity of antioxidant enzymes, lowering the levels of oxidative stress indicators, and regulating the Nrf2/HO-1 pathway [150,152]. These findings were validated through the observation of histological damage in the colon, where a reduction in inflammation and the mitigation of chronic colitis lesions could be seen [151,152,153,155]. An improvement in the abundance and variety of gut bacteria was detected, indicating a reduction in gut dysbiosis [156]. Furthermore, the integrity of the gut’s protective mucosal barrier was restored and intestinal innate immunity was enhanced, which led to the overall improvement of intestinal health [157]. These findings indicate that isolated compounds derived from macroalgae may hold significant promise for the treatment of gut diseases, whereas intestinal health was mainly improved by hydroquinones [158] and polysaccharides [156]; (ulcerative) colitis by terpenes [155], phlorotannins [152,154], and carotenoids [151,152,154]; and hepatic health by polysaccharides [146] and polyphenols [147]. Interestingly, a commercial formulation composed of polysaccharides, phlorotannins, and other polyphenols [145] displayed promising results for NASH and NAFLD, which may indicate that these compounds could have synergistic effects.

**Table 8 molecules-29-01900-t008:** In vivo studies regarding algal extracts/compounds with gastrointestinal implications.

Complication	Algae Type	Algae Species	Algal Extraction or Compound	Route of Administration	Dosage	Experimental Period	Animal Model (Age)	Induced by	*n*/Group	Outcomes and Mechanism	References
Hepatic damage	Macroalgae	*Gracilaria caudata*	Sulfated polysaccharides	Intraperitoneally	10 mg/kg/day	5 d	Swiss mice	Nimesulide (intragastric)	8 to 10	↓ liver weight; improved antioxidant parameter in liver (↓ MPO, MDA, and NO3/NO2 levels, ↑ GSH level); ↓ inflammatory markers (↓ IL-1β and TNF-α levels); enhancement of hepatic function markers (↓ AST, ALT, and GGT levels)	[146]
Liver fibrosis	Macroalgae	*Caulerpa racemosa*	Water extracts	Oral in distilled water	200 mg/kg/mL	5 w	Wistar Rats	40% Carbon tetrachloride (CCl_4_) intraperitoneally	7	Enhanced liver enzymes in serum (↓ ALT, AST, ALP, and LDH) and liver metabolite (↓ total bilirubin and direct bilirubin); improved renal and lipid profile (↓ urea; ↓ creatine); increased hepatic antioxidant enzymes (↑ GSH and CAT; ↓ MDA)	[148]
		*Padina pavonia*							6		
NAFLD	Macroalgae	*Ishige okamurae*	Diphlorethohydroxycarmalol (DPHC)	Incubation with embryo media	40 µM	3 d	Transgenic zebrafish embryos (Danio rerio) (3 dpf)	Palmitate	12 to 15	↓ lipogenesis (downregulation of lipogenesis-related genes SREBP1c, ChREBP1 α, and FAS); ↓ liver inflammation (↓ IL-1β, TNF-α, and COX-2); regulation of lipid metabolism (stimulation of AMPK and SIRT1 signaling pathway)	[147]
NASH and NAFLD	Macroalgae	*A. nodosum* and *F. vesiculosus*	Gdue©	Intragastric gavage	7.5 mg/kg bw	12 w for NAFLD; 18 w for NASH	Sprague Dawley rats (4–8 w)	Western diet high-fat diet plus 30% fructose in the drinking water	10	↓ body weight gain; ↓ plasma glucose; reduced liver steatosis (↓ liver TG accumulation); ↓ hepatic inflammation; restored hepatic lipid metabolism (downregulation of lipid droplet forming and fatty acid synthase genes); restored physiological levels of protein expression regulating lipid homeostasis	[145]
IBD	Macroalgae	*Ulva pertusa*	Extract	Oral gavage	50 mg/kg and 100 mg/kg	4 d	CD1 mice (4 w)	DNBS injected into the rectum	10	↓ body weight loss; ↑ pain threshold; ↓ DNBS-induced hyperalgesia; ↓ DNBS-induced visceral hypersensitivity; ↓ cell adhesion molecules (↓ ICAM-1 and P-selectin); ↓ gut-inflammation (reduced IL-6, IL-17, and IL-23 levels and enhanced serum IL-10 levels); modulation of innate (↓ CD68^+^ cells) and adaptive (↓ CD4^+^ and CD8^+^ cells) immune system; TLR4 and NLRP3 inflammasome modulation	[149]
									11	↓ body weight loss; ↓ inflammation (reduced the expression levels of NF-κB and restored the expression of Ikb-α); modulation of pro-inflammatory interleukin production (↓ IL-5, IL-9, and IL-13, and ↑ IL-4); modulation of apoptosis (↓ p-53, caspase-3, -8, and -9, and ↓ ↑ Bcl-2); ↓ oxidative stress (↓ MDA levels, ↑ GSH, CAT, SOD, Mn-SOD, and HO-1); modulation of Nrf2/SIRT1 pathway (↑ Nrf2 and SIRT1 levels)	[150]
Intestinal Health	Macroalgae	*Cymopolia barbata*	Cymopol	Oral gavage	0.1 g/kg and 0.4 g/kg	3 d	C57BL/6J mice (4 w)	3% DSS in drinking water	5	↓ inflammatory and oxidant response (downregulation of ERK/MAPK and PI3K/AKT pathways)	[158]
		*Laminaria* spp.	Enzymolysis seaweed powder	Feed	20 g/kg of feed	4 m	Ragdoll kittens (6 m)		10	↑ growth performance (↑ weight gain); improved immune function (↑ IgG and IgA); improved antioxidant parameters (↑ SOD, ↓ MDA); ↓ inflammation (↓ IL-1β, IL-6, and TNF-α, and ↑ IL-10); more microbiota richness and diversity (↑ relative abundance of Bacteroidetes, Lachnospiraceae, Prevotellaceae, and Faecalibacterium); improved gut mucosal barrier function	[157]
		*Porphyra yezoensis*	Oligoporphyran	Fish meal	1%	8 w	Adult zebrafish (1 m)	n.d.	25	Positive effect on digestive enzymes (Protease, Lipase, and Amylase) activity; enhanced lipid content of body composition; enhanced intestinal innate immunity (↑ lysozyme); ↓ inflammation in intestines (↑ IL-10); improvement in gut microbial community	[156]
Colitis	Macroalgae	*Laurencia glandulifera*	O11,15-Cyclo-14-Bromo-14,15-Dihydrorogiol-3,11-Diol	Intraperitoneally	0.25 mg/mouse every 48 h	5 d	C57BL/6	DSS in drinking water	3 to 5	↓ Inflammation (↓ IL-1β, TNF-α, and IL-6)	[155]
			Neorogioldio								
		n.d.	Eckol	Oral gavage	1 mg/kg	3 w	C57BL/6J (7–8 w)	DSS	15	↓ body weight loss; attenuation of colitis symptom; improvement in colon shortening; ↓ pro-inflammatory cytokines in colon (TNF-α, IL-1β, and IL-6, ↑ IL-10); downregulation of NF-κB and TLR4 in colons; ↓ apoptosis (↓ caspase-9 protein expression); improved gut microbiota dysbiosis; immunoregulatory effect in colitis (recruitment of dendritic cells to the colonic tissue)	[154]
UC	Macroalgae	*Turbinaria ornate*	Methanol fraction from ethanol extract	Oral	15 mg/kg/	6 w	C57BL/6J mice (7 w)	DSS	6	↓ inflammatory response (↓ MPO activity, ↓ COX-2, p-STAT-3, and TNF-α expression levels, ↑ IL-10 and FOXP3 expression levels); upregulation of regulatory T cell activity	[153]
		n.d.	Fucoxanthin	Oral	50 mg/kg/day and 100 mg/kg/day	2 w	C57BL/6J (8 w)	DSS in drinking water	10	↓ body weight loss; improved colon shortening; ↓ inflammation in colon tissues (prevention of increase in colonic PGE2 production, ↓ COX-2 expression and ↓ NF-κB activation)	[151]
	n.d.	n.d.	Dieckol	Oral gavage	5, 10, and 15 mg/kg	11 d	C57BL/6J mice	DSS (3% in drinking water)	6	↓ body weight loss; ↑ colon length; ↓ oxidative stress mediators (↓ MPO and MDA activity) in colon tissue; ↓ inflammation (↓ COX-2, IL-1β, and TNF-α); NF-κB inhibition and upregulation of Nrf2/HO-1 signaling cascade	[152]

AKT: Protein kinase B; ALP: Alkaline phosphatase; ALT: Alanine aminotransferase; CAT: Catalase; ChREBP1: Carbohydrate-responsive element-binding protein 1; COX: Cyclooxygenase; d: Days; DSS: Dextran sulfate sodium; ERK: Extracellular signal-regulated kinase; FAS: Fatty acid synthase; GGT: Gamma-glutamyl transferase; GSH: Glutathione; HO-1: Heme oxygenase-1; IBD: Inflammatory bowel disease; Ig: Immunoglobulin; IL: Interleukin; m: Months; MDA: Malondialdehyde; Mn-SOD: Manganese superoxide dismutase; MPO: Myeloperoxidase; NAFLD: Non-alcoholic fatty liver disease; NASH: Non-alcoholic steatohepatitis; NLRP3: NOD-, LRR-, and pyrin domain-containing protein 3; NF-κB: Nuclear factor-kappa B; n.d.: No data available; Nrf2: Nuclear factor erythroid 2-related factor 2; PGE: Prostaglandin E; PI3K: Phosphoinositide 3-kinase; ROS: Reactive oxygen species; SIRT1: Sirtuin 1; SOD: Superoxide dismutase; SREBP1c: Sterol regulatory element-binding protein 1c; TNF: Tumor necrosis factor; TLR: Toll-like receptor; UC: Ulcerative colitis; w: Weeks; ↑: Increase; ↓: Decrease.

### 3.5. Metabolic Health-Promoting Activity

#### 3.5.1. Obesity, Diabetes, and Metabolic Health

Obesity is a chronic and very frequent condition that is anticipated to afflict 20% of the global population by 2025, with its hallmarks being dysfunctional adipose tissue and chronic low-grade inflammation [16]. Adipocyte enlargement leads to oxidative stress, the increased production of endothelial adhesion molecules, and the release of pro-inflammatory mediators (mainly through the activation of the JNK and NF-κB signaling pathways), that lead to the infiltration and activation of pro-inflammatory immune cells, thereby intensifying local and systemic inflammation through a positive feedback loop [159]. The combination of chronic inflammation and dietary factors, specifically the consumption of a high-fat diet, leads to an imbalance in the gut microbiota. The higher fat consumption increases mitochondrial stress, leading to dysfunctions inside intestinal epithelial cells, deteriorating their ability to maintain anaerobiosis-driven gut homeostasis and shifting gut microflora from obligate to facultative anaerobes. As discussed previously, dysbiosis in the gut microbiota can lead to oxidative stress, inflammation, the buildup of toxic metabolites, the impaired breakdown of SCFAs, and aberrant immune responses. All these factors contribute to the development of diabetes and metabolic syndrome risk factors such as insulin resistance [16,139].

Diabetes Mellitus type 2 is a chronic disease characterized by hyperglycemia, which is frequently a result of the body’s impaired production of insulin from pancreatic beta-cells or the diminished ability of target tissues (such as skeletal muscle, liver, and adipose tissue) to respond effectively to insulin. Insulin resistance typically triggers an excessive release of insulin by beta-cells, which further decreases the insulin sensitivity of tissues, leading to persisting hyperglycemia over time. Obesity-induced insulin resistance is mainly caused by chronic inflammation (mediated by the JNK and NF-κB pathways) and oxidative stress originated in the adipose tissue. These factors hinder the insulin signaling pathway, specifically PI3K/PKB, resulting in impaired glucose uptake (mainly by preventing the translocation of glucose transporter proteins) and glycogen storage. Certain cytokines can additionally induce apoptosis in beta-cells by activating the MAPK and NF-κB pathways [159]. Gut dysbiosis also seems to play a role in causing diabetes and insulin resistance [139]. Insulin resistance causes microvascular damage, mainly in the heart, blood vessels, eyes, kidneys, and nerves, which can lead to hypertension, renal impairment, ischemic heart disease, metabolic cardiac inflammation, and diabetic retinopathy, among many others [160].

Abdominal obesity and insulin resistance, together with dyslipidemia and hypertension, are primary risk factors for the development of other NCDs. They are considered crucial drivers of the current global cardiovascular crisis by creating a favorable environment for the development and aggravation of insulin resistance and inflammation. This sets the stage for the development of more disorders that compromise overall metabolic health. The occurrence of these illnesses, in turn, heightens the likelihood of developing metabolic syndrome [161].

Obesity and diabetes involve dysfunctions in carbohydrate and lipid metabolism; therefore, some therapeutic agents target digestion enzymes to delay or block the absorption of these macronutrients. Inhibiting amylase and glucosidase, which break carbohydrates into glucose monomers, slows glucose absorption and prevents post-prandial hyperglycemia. Other anti-diabetic strategies include inhibiting enzymes involved in insulin signaling or secretion, such as protein tyrosine phosphatase 1B (PTP-1B) and dipeptidyl peptidase-4 (DPP-4). Inhibiting these enzymes improves insulin signaling and sensitivity (PTP-1B) and insulin secretion (DPP-4), which improves glycemic control in diabetes [162]. One approach to managing obesity involves the use of therapeutics that specifically target lipase, the enzyme that breaks down dietary fats in the gastrointestinal tract. By inhibiting the lipase activity, fat digestion and absorption are decreased while also decreasing overall caloric intake [163].

#### 3.5.2. Algal Compounds in Weight Control and Metabolism Regulation

During this literature review, only three obesity-related studies conducted in vitro were identified, all of which focused on the use of seaweed extracts in the Lipase Inhibition Assay. In the work of Kurniawan et al. (2023) [84], three fractions (ethanol, hexane, and ethyl acetate) of *Caulerpa racemosa* were analyzed, with IC_50_ values of 45.5, 48.0, and 59.1 µg/mL, respectively. In the study of Nurkolis et al. (2023) [164], the maceration hexane extract of seaweed *Caulerpa lentillifera* showed an anti-obesity potential via in vitro Lipase Inhibitory Assay which was stronger than that of orlistat, a standard drug used in the management of obesity, (92.1 vs. 95.2 µg/mL). In the study of Catarino et al. (2019) [165], the acetone extract of *Fucus vesiculosus* presented an IC_50_ value of 45.9 µg/mL, whereas its ethyl acetate fraction showed a significantly lower value of 19.0 µg/mL.

The in vitro assays (Table 9) regarding anti-diabetic potential (*n* = 16) showed the inhibition of four different types of enzymes involved in the glucose digestion (α-amylase and glucosidase) and metabolism (DPP-4 and PTP-1B). All studies concerned macroalgae, but whereas most studies investigated these activities of extracts, only in the case of *Padina tetrastromatica* [166], *Hydropuntia edulis* [167], and *Turbinaria ornate* [168] were isolated compounds tested (dolabellanes and dolastanes, sulfated pyruvylated polysaccharide, and turbinafuranones, respectively). All of these compounds exhibited comparable efficacy to the anti-diabetic reference compounds. The dolabellanes and dolastenes demonstrated antidiabetic potential by effectively inhibiting α-amylase and α-glucosidase, comparable to acarbose (with an IC_50_ value of 120–140 µg/mL). Additionally, the polysaccharide and turbinafurones demonstrated the inhibition of DPP-4 and PTP-1B, comparable to the standard inhibitors diprotin-A (with an IC_50_ of 4.21 µM) and sodium metavanadate (with an IC_50_ of 2.52 mM), respectively. The antidiabetic potential of water extracts from *Pterocladia capillacea, Sphacelaria rigidula*, and *Stoechospermum marginatum* was mainly attributed to their phenolic content [169]. In the case of *Fucus vesiculosus*, the bioactive compounds were found to be more concentrated in the ethyl acetate fraction of the acetone extract, as this fraction showed a ten-fold increase in α-amylase inhibition and a five-fold increase in α-glucosidase compared to the crude extract [165]. This is mainly attributed to the fraction’s higher content in phlorotannins, which are known for their anti-diabetic activity. The ethyl acetate fraction of methanol extract from *Halimeda tuna* displayed promising results in the α-glucosidase inhibitory activity, being mainly attributed to alkaloids, flavonoids, steroids, and phenol hydroquinone [170].

Only one cell assay was found to fulfil the inclusion criteria for this review regarding a complication of diabetes, diabetic retinopathy [179]. Fucoxanthin (0.1 and 0.5 mg/mL), of not-described origin was incubated with ARPE-19 cells stimulated with high lipid peroxidation 4-hydroxynonenal and high glucose condition, and showed reduced cell viability, cell and DNA damage, morphology changes, and apoptosis. Additionally, fucoxanthin maintained the integrity of the blood–retinal barrier and decreased oxidative stress by decreasing ROS and increasing CAT activity, therefore showing promising results for the mitigation of diabetic retinopathy.

A total of 14 in vivo trials were found, with 13 involving the effect of macroalgal compounds and only 1 involving a compound of microalgal origin (Table 10). Algal compounds were able to modify all the risk variables associated with metabolic syndrome: obesity, insulin resistance, inflammation, and oxidative stress. Obesity was reduced by decreasing weight and fat gain [180,181,182,183] and restoring the gut microbiota (including an increase in the production of SCFAs) [183,184]. Insulin resistance was improved by reducing plasma glucose levels [180,185] and α-amylase and glucosidase activity [186], increasing glucose transport [187], decreasing serum insulin levels [181], and enhancing insulin sensitivity as well as improving beta-cell function [188]. Additionally, the phlorotannin extract of *Cystoseira compressa* improved the islet size and function; restored necrotic and fibrotic changes; and reduced the number of degenerative cells in islets [186]. Inflammation and oxidative stress [189] were reduced by decreasing the production of pro-inflammatory markers and modulating the inflammatory signaling pathways [182]. Furthermore, there was a mitigation of complications typically associated with obesity and diabetes, such as improvements in cardiovascular health [182,183] (lower blood pressure, reduced cardiac inflammation, decreased collagen deposition in the heart, and the recovered morphology of cardiac tissues), kidney and liver health [181] (decreased levels of liver and kidney parameters in the blood, indicating less damage), and improved dyslipidemia (lower total cholesterol levels in the blood) [181,182,184].

Based on the accumulated evidence, it seems that several extracts, fractions, and isolated compounds (chlorophyll catabolite [190], phlorotannin [187], polyphenol [188], and polysaccharides [182,184]) obtained from algae hold promising potential in reducing the symptoms and complications of diabetes and obesity. Furthermore, they can enhance metabolic health, reducing the risk of developing metabolic syndrome and providing a way out of the harmful cycle of non-communicable diseases.

**Table 10 molecules-29-01900-t010:** In vivo studies regarding algal extracts/compounds with metabolic benefits.

Algae Type	Algae Species	Algal Extraction or Compound	Route of Administration	Dosage	Experimental Period	Animal Model (Age)	Induced by	*n*/Group	Outcomes and Mechanism	References
Macroalgae	*Caulerpa lentillifera*	Biomass	Feed	5%	16 w	Wistar rats (8–9 w)	High-carbohydrate, HFD	12	↓ body weight gain; ↓ fat gain (↓ retroperitoneal, epididymal, omental, total abdominal, and visceral fat, and adiposity); ↓ systolic blood pressure; ↓ lipids (↓ TC and liver fat vacuole area); modulation of gut bacteria (↓ Firmicutes to Bacteroidetes ratio)	[183]
	*Caulerpa racemosa*	Ethyl acetate extract	Oral-feeding tube	100 and 200 mg/kg body weight	24 d	Sprague Dawley rats (8 w)	STZ (i.p.)	6	↓ plasma glucose level; ↓ ALT and AST levels (plasma); ↑ Albumin levels	[185]
	*Cystoseira compressa*	Phlorotannin extracts	Oral	60 mg/kg	6 w	Wistar albino rats	STZ (i.p.)	10	↓ blood glucose, ↓ α- amylase and ↓ glucosidase activity; ↓ urea; ↓ creatine; ↓ oxidate stress (↑ GSH and CAT; ↓ MDA)	[186]
	*Dictyota dichotoma*	n-butanol and ethyl acetate extracts	n.d.	100 and 200 mg/kg	3 d	Rats	Monohydrate (i.p.)	6	Hypoglycemic activity (↓ blood glucose level); activation of AMPK pathway	[191]
	*Gelidium amansii*	Pheophorbide A (PhA)	Oral	10 mg/kg	3 w	ICR mice (4 w)	STZ injection	7	↓ blood glucose after 30, 60, and 120 min; ↓ postprandial blood glucose levels	[190]
	*Ishige okamurae*	Diphlorethohydroxycarmalol	Injection	0.3 µg/g body weight	90 min	Wild-type zebrafish (adults)	Alloxan (2 mg/mL) and glucose (1%)	3	↓ blood glucose levels; ↑ glucose transport (↑ calcium levels in skeletal myotubes and ↑ Glut4 translocation and ↑ phosphorylation of AMPK); regulation of muscle contraction	[187]
	*Palmaria palmata*	Alcalase/Flavourzyme-produced protein hydrolysate	Oral gavage	100 mg/kg	180 min	NIH Swiss mice (10–12 w)	Glucose	8	Improved glucose tolerance (↓ blood glucose level)	[192]
	*Polysiphonia japonica*	5-Bromoprotocatechualdehyde	Incubation with embryo media	50 µM	35 h	Zebrafish embryos (3 dpf)	Palmitic acid (0.2 mM); stimulation with glucose	10 to 12	Protective effect against PA-induced β-cells dysfunction (↑ insulin secreting cells)	[188]
	*Rhodomela confervoides*	3,4-Dibromo-5-(2-bromo-6-(ethoxymethyl)-3,4- dihydroxybenzyl)benzene-1,2-diol (BPN)	Oral gavage	20 mg/kg	12 w	Wistar Rats	Diet induced obesity; STZ (i.p.)	10	↓ blood glucose	[193]
	*Sargassum pallidum*	Fucoidan	Intragastric	200 mg/(kg/d)	8 w	C57BLKS/J db/m and db/db mice (7–8 w)- spontaneous diabetic model	-	6	↓ weight gain; ↓ hyperlipidemia (↓ TG and TC); anti-diabetic activity (↑ glucose tolerance, ↓ insulin resistance, and ↑ insulin sensitivity); ↓ oxidative stress on cardiac tissue (↓ MDA in serum and heart and ↑ SOD, CAT, and GSH/GSSG, ↓ lipid peroxidation); counteracted the repression of AMPK/Nrf2/ARE antioxidant signaling axis in cardiac tissue; ↓ hyperglycemia-associated metabolic cardiac inflammation (↓ activation of NF-κB signaling pathway and ↓ mRNA levels of Il-1β, Il-6, and TNF-α)	[182]
	*Saccharina japonica*	Dietary fibers	Oral	500 mg/kg/day	9 w	C57BL/6JGpt (4 w)	HFD	12	↓ body weight; ↑ insulin sensitivity; ameliorated dyslipidemia (↓ TG, LDL-c, and FFA and ↓ visceral fat index); alleviated liver (↑ ALT and AST) and renal (↓ creatine and urea) damage; antioxidant effect on liver (↓ MDA, ↑ CAT, GSH, and SOD); anti-inflammatory potential (↓ TNF-α, IL-6, and MCP-1); improved gut microbiota dysbiosis; modulation of SCFA metabolism (increase in SCFAs production in colonic contents)	[184]
	*Ulva reticulata*	Methanol extract	Oral	250 mg/kg	31 d	Wistar albino Rats (2–3 m)	STZ injection	6	↓ cholesterol, ALT, TG, and AST; ↓ blood glucose level; ↓ body weight	[181]
		Chloroform fraction	Oral	10 mg/kg	17 d			7	↓ cholesterol, ALT, TG, and AST; ↓ blood glucose level; ↓ body weight; ↓ serum insulin	
	n.d.	Fucoidan	Intraperitoneal	100 mg/kg	6 w	Wistar albino Rats (3 m)	STZ injection	9	↓ blood glucose; ↓ body weight	[180]
Microalgae	*Tetraselmis chui*	TetraSOD^®^	Oral	17 mg/kg body weight	16 w	Sprague–Dawley (7 w)	Diet-induced obesity by cafeteria diet	10	↓ oxidative stress (↑ Nox) levels and anti-inflammatory markers (↓ IL-10); ↑ antioxidant enzymes in liver (↑ GSH); modulation of genes involved in antioxidant and anti-inflammatory pathways in the liver, mesenteric white adipose tissue, spleen, and thymus.	[189]

AMPK: AMP-activated protein kinase; AST: Aspartate aminotransferase; Bax: Bcl-2-associated X protein; Bcl: B-cell lymphoma; CAT: Catalase; dpf: Days post fertilization; GSH: Glutathione; GSSG: Glutathione disulfide; HFD: High-fat diet; i.p.: Intraperitoneally; IL: Interleukin; m: MonthsMDA: Malondialdehyde; n.d.: No data available; Nrf2: Nuclear factor erythroid 2-related factor 2; PARP: Poly(ADP-ribose) polymerase; ROS: Reactive oxygen species; SCFA: Short-chain fatty acid; SOD: Superoxide dismutase; STZ: Streptozotocin; TC: Total cholesterol; TG: Triglycerides; TNF: Tumor necrosis factor; w: Weeks; ↑: Increase; ↓: Decrease.

### 3.6. Anti-Cancer Activity

#### 3.6.1. Cancer Development and Progression

Cancer is an umbrella term for diseases characterized by uncontrolled cell growth, with the potential to spread to different regions of the body through metastasis, being the second cause of mortality worldwide and a major limiting factor for increased life expectancy. The most prevalent types of cancer in men include lung, prostate, colorectal, stomach, and liver cancer. In women, the most common types are breast, colorectal, lung, cervical, and thyroid cancer [194]. Currently, the main therapeutic approaches for cancer are surgery, radiotherapy, immunotherapy, and chemotherapy. However, these approaches usually possess low selectivity and attack both tumoral and healthy cells. Other approaches include photodynamic therapy, hyperthermia, non-traditional therapy with natural bioactive materials, and tumor vaccination [195]. There are six main hallmarks of cancer: uncontrolled proliferation; evading growth suppressors; enabling replicative immortality; invasiveness and metastasis; inducing angiogenesis; and resisting cell death [195].

#### 3.6.2. Oxidative Stress and Inflammation in Cancer Development

Both oxidative stress and chronic inflammation can be considered enabling characteristics as they contribute to DNA damage and aberrant signaling pathways, creating a tumor-promoting microenvironment [195,196]. Cancerous cells have a higher ROS production than healthy cells due to their uncontrolled proliferation, which increases mitochondrial activity and creates a hypoxic environment due to the insufficient blood supply. In cancer cells, mitochondrial-generated and NOx-generated ROS promote pro-tumorigenic signaling and proliferation while, simultaneously, aerobic glycolysis is upregulated. This leads to increased DNA damage, more production of ROS, and the depletion of oxidative defense enzymes. However, paradoxically, cancerous cells have adapted to higher ROS levels and developed strategies to keep ROS in a tolerable and functional range, avoiding cell death [197,198]. This includes the upregulation of the cells’ antioxidant defense system (namely, through the upregulation of Nrf2). Interestingly, ROS can have pro- or anti- tumorigenic functions, depending on ROS levels. If ROS levels are only moderately elevated in cancerous cells, there is an upregulation of the MAPK/ERK and AKT/PIK3/mTOR pathways, which promote cell proliferation and inactivate pro-apoptotic factors, therefore supporting the survival, chronic inflammation, angiogenesis, and metastasis of cancerous cells. Additionally, a moderate increase in ROS in cancer cells seems to be associated with the activation of oncogenes, the inactivation of tumor suppressor genes, the promotion of angiogenesis, and mitochondrial dysfunction. On the other hand, if ROS levels are above the tolerable threshold, they can lead to cell death, through the apoptosis, autophagy, or necroptosis of the cancer cells. Therefore, some therapies rely on either significantly increasing ROS levels in cancerous cells or reducing ROS to normal levels to stop the proliferation of cancerous cells; both mechanisms are under investigation [195,196,197,198]. Cancer cells also have mechanisms of immune evasion, which ROS can modulate, that either has immunostimulatory or immunosuppressive activity. In some ROS-inducing treatments, an adaptive anti-cancer immune response has been seen, where protection against the tumor is mediated via the development of antitumoral immunity. Ideally, both mechanisms (anti-proliferative) and immune stimulatory activities act synergistically, allowing for better patient outcomes [195,196,197,198].

Chronic inflammation, another hallmark of cancer, is related to almost 20% of human cancers by enabling oncogenic mutations and creating a tumor-promoting microenvironment through the activation of inflammatory signaling pathways, such as NF-κB and MAPKs. This promotes angiogenesis, supporting the growth and survival of cancer cells [195]. Additionally, inflammation is responsible for the release of cytokines and, under hypoxic conditions, growth factors such as vascular endothelial growth factor (VEGF), which are vital for the process of angiogenesis, therefore promoting the vascularization and rapid expansion of tumors. Inflammation is also involved in the remodulation of epithelial–mesenchymal tissue, mainly through the upregulation of matrix metalloproteinases (MMPs), which are known to degrade the proteins found in the basal membrane. In homeostasis conditions, this degradation aids in the migration of immune cells; however, in cancer, the dysregulation of cell adhesion mediated by MMP activity is often associated with invasion and metastasis [195,196,197,198]. Although tumorigenesis is usually related to the site of the chronic inflammation (e.g., colitis, inflammatory bowel disease, pancreatitis, and hepatitis, are linked to a greater risk of colon, colorectal, pancreatic, and liver cancers, respectively), if inflammation is systemic, metastasis might occur, and other body parts might be affected. In general, as inflammation is linked to all stages of tumorigenesis, the inhibition of chronic inflammation is regarded as beneficial and several anti-cancer therapeutics have anti-inflammatory effects [50].

#### 3.6.3. Algal Compounds with Anti-Cancer Properties

A total of 26 cell experiments from 23 studies (Table 11) fulfilled the inclusion criteria defined for this current review, with the five main cancer categories being “breast cancer” (*n* = 5), “colorectal cancer” (*n* = 3), “liver cancer” (*n* = 4), “lung carcinoma” (*n* = 3), and skin-related cancer (*n* = 2), which coincide with the most prevalent cancer types. Interestingly, anti-cancer activity featured the highest amount of microalgal studies (*n* = 7) compared to all the other bioactivities described in this review, which confirms it being a research hotspot [199]. In general, the tested macroalgal and microalgal extracts/compounds were successful in targeting one or multiple hallmarks of cancer—by displaying anti-proliferative effects (mainly through the induction of apoptosis) [72,200,201,202,203,204,205,206,207,208,209,210,211,212,213,214], decreasing the inflammatory state [200,211,212,213], migration capacity [215], angiogenesis potential [213], and adhesion properties [213]. When modulating oxidative stress, mechanisms to decrease cancerous cell survival differed between extracts or isolated compounds. The methanol extract of *Skeletonema marinoi* [205] led to a decrease in NOx-generated ROS, increased antioxidant enzyme production, and decreased DNA damage in leukemia cells. On the other hand, the purified compounds triphlorethol-A [200] and fucoidan [210] increased oxidative stress by reducing antioxidant enzyme production, downregulating antioxidant pathways, and increasing DNA damage in brain glioma and oral cancer cells, respectively. However, both strategies were successful in decreasing cancer cell proliferation and survival, adding to the debate of therapeutic approaches towards ROS.

In addition to their anti-proliferative effects, polysaccharides from two macroalgae (*Chondrus armatus* [214] and *Sargassum pallidum* [204]) also exhibited immunotropic activity, potentially enhancing their overall effectiveness.

Some of the tested compounds were also evaluated for their selective cytotoxicity, in which healthy cells were left unharmed. A study investigating the use of fucoxanthin for breast cancer treatment found that combining this compound with other already established therapies increased its selectivity, suggesting significant potential for its use in combination therapy [202]. The subfractions of *Tisochrysis lutea* dichloromethane extract [216] also displayed high selectivity, maybe due to identified carotenoid-derived metabolites—loliolide and epi-loliolide. Regarding breast cancer, the anti-proliferative and cytotoxic activities (e.g., by decreasing cell viability and inducing morphological changes in cancerous cells) of *Chaetomorpha* sp. [217] and *Nannochloropsis oculate* [218] extracts may be linked to the high content of dichloracetic acid, oximes, and L-a-Terpinol [217], and terpenoids, carotenoids, polyphenolic, and fatty acids [218] in their respective extracts.

**Table 11 molecules-29-01900-t011:** Cell experiment studies regarding algal extracts/compounds with anti-cancerous activity.

Complication	Algae Type	Algae Species	Algal Extract or Compound	Cell Line	IC_50_ of Cell Viability	Concentrations Tested	Outcomes and Mechanism	Reference
Brain glioma	Macroalgae	*Ecklonia cava*	Triphlorethol-A	U251	20 µM		↑ ROS accumulation; ↑ mitochondrial apoptosis (↑ chromatin condensation, fragmented nuclei, and membrane blebbing); ↓ antioxidant enzymes (SOD, CAT, and GSH); ↑ Bax expression and ↓ Bcl-2; ↑ protein expression of cytochrome *c* and caspase-3 and -9; downregulation on phosphorylated JAK2/STAT3 and MAPK/ERK1/2 pathways	[200]
Breast Cancer	Macroalgae	*Caulerpa racemosa*	Crude polyphenolic extract	KAIMR C1	168.5 µM			[172]
		*Chaetomorpha* sp.	Ethanol extract	MDA-MB-231	225.2 µg/mL			[217]
		*Sargassum myriocystum*	Methanol extract	MCF-7	66.8 µg/mL		↓ cell viability (morphological cell changes indicating apoptosis)	[201]
	Microalgae	*Nannochloropsis oculata*	Methanol extract	MDA-MB-231		200, 400, and 600 µg/mL	↓ cell viability; morphological changes in cancerous cells	[218]
	n.d.	n.d.	Fucoxanthin	MDA-MB-231; MCF-1; SKBR3		10 µM	↓ cell viability tumoral cell lines; (↑ apoptotic cells, ↓ cell proliferation, and ↑ cell damage) when used in combination with known anti-cancer drugs (cisplatin and doxorubicin)	[202]
Colon Cancer	Macroalgae	*Caulerpa racemosa*	Crude polyphenolic extract	HCT-8	160.0 µM			[172]
		*Laurencia synderiae*	Methanolic extract	HT-29	70.2 µg/mL	50–100 µg/mL	↑ cell death of cancerous cells (↓ cell viability); ↑ apoptosis (↑ chromatin condensation, nuclear fragmentation, and DNA fragmentation)	[203]
Colorectal cancer	Macroalgae	*Ecklonia maxima*	Fucoidans	HCT-116		0.1–0.5 mg/mL	↓ cell adhesion; ↓ colony formation; ↓ cancer cell sphere formation; ↓ cancerous cell migration 2D and 3D models	[215]
		*Ecklonia radiata*						
		*Sargassum elegans*						
Esophageal adenocarcinoma	Macroalgae	*Chondrus armatus*	Carrageenans	FLO1		100 and 400 µg/mL	↓ cell viability of cancerous cells	[214]
Gastric cancer	Microalgae	*Gloeothece* sp.	Hexane–isopropanol extract	AGS	23.2 µg/mL		anti-proliferative effect (↑ cell death of cancerous cells); not toxic in non-cancerous cells (HCMEC cell line) up to 100 µg/mL	[72]
Hepatic cancer	Macroalgae	*Sargassum pallidum*	Polysaccharide fractions	HepG2		25, 100, and 400 µg/mL	↓ cell viability of cancerous cells (↑ apoptosis)	[204]
Leukemia	Microalgae	*Skeletonema marinoi*	Methanol extract	K562		0.75 mg/mL	↓ cell viability; ↑ cell apoptosis (↑ proapoptotic Bax protein expression and ↓ antiapoptotic protein Bcl-2 expression); ↓ oxidative stress (↓ NO production through NOX2 pathway); restored redox status (↑ SOD, CAT, and GPx); ↓ oxidative DNA damage	[205]
Liver cancer	Macroalgae	*Dictyotaciliolata*	Methanol or aqueous extracts	HepG2		0.05–1 mg/mL	↓ cell viability by inducing apoptosis (↑ caspase 3 and 9 activity)	[206]
		*Pyropia Yezoensis*	Phycoerythrin	HepG2		20 and 30 µg/mL	↓ cell viability (altered cell membrane integrity; ↑ apoptosis)	[207]
	Microalgae	*Tisochrysis lutea*	Dichloromethane extract	HepG2	85.1 µg/mL		Subfractions displayed high selectivity index (S17 vs. HepG2).	[216]
Lung carcinoma	Macroalgae	*Sargassum pallidum*	Polysaccharide fractions	A549		25, 100, and 400 µg/mL	↓ cell viability of cancerous cells (↑ apoptosis)	[204]
		*Udotea flabellum*	Hydrolysated protein	A549	300.7 mg/mL			[128]
	Microalgae	*Oscillatoria simplicissima*	Sulfated polysaccharides	A549		100 µg/mL	↓ cell viability of cancerous cells	[219]
Melanoma	Microalgae	*Isochrysis galbana*	Methanol extract and fractions	A2058		100 µg/mL	Antiproliferative effect of cancerous cells (↓ cell viability)	[208]
Nasopharyngeal carcinoma	n.d.	n.d.	Fucoxanthin	C666-1		25 µM	Cytotoxic effect by inducing autophagy and apoptosis	[209]
Oral cancer	n.d.	n.d.	Fucoidan	Ca9-22 CAL 27		800 and 1200 µg/mL	Selectively cytotoxic to cancer cells but not in non-malignant oral cells; ↑ apoptosis in cancerous cells (↑ activation of caspase-8, -9, and -3); ↑ ROS levels in oral cancer cells (downregulation of antioxidant signaling genes NRF2, TXN, and HMOX1); DNA damage-inducible effects in cancer cells	[210]
Ovarian cancer	Macroalgae	*Agarum clathratum*	Extract	ES2 and OV90		25 µg/mL	↓ cell viability (induced apoptosis; ↓ phosphorylation of ERk1/2 MAPK)	[211]
Pancreatic cancer	Macroalgae	*Ecklonia cava*	Dieckol	PANC-1	20 μM		↑ apoptosis (↓ Bcl2 expression and ↑ Bax); ↑ ROS generation in cancerous cells; ↓ antioxidant enzymes (SOD, CAT, and GSH); ↓ cell adhesion; anti-inflammatory activity (↓ TNF-α, IL-6, IL-8, and IL-1β)	[212]
Prostate cancer	Microalgae	*Skeletonema marinoi*	Methanol extract	DU145 LNCaP		100 µg/mL	↓ cancerous cell proliferation; ↑ apoptosis; ↓ cell vascular mimicry; downregulation of inflammation- and angiogenesis-associated genes	[213]
Squamous-cell carcinoma	Macroalgae	*Chondrus armatus*	Carrageenans	KYSE30		100 and 400 µg/mL	↓ cell viability of cancerous cells	[214]

Bax: Bcl-2-associated X protein; Bcl: B-cell lymphoma; CAT: Catalase DNA: Deoxyribonucleic acid; EAC: Ehrlich ascites carcinoma; ERK: Extracellular signal-regulated kinase; GPx: Glutathione peroxidase; GSH: Glutathione; IL: Interleukin; JAK: Janus kinase; MAPK: Mitogen-activated protein kinase; NO: Nitric oxide; NOX2: NADPH oxidase 2; n.d.: No data; ROS: Reactive oxygen species; SOD: Superoxide dismutase; STAT: Signal transducer and activator of transcription; TNF: Tumor necrosis factor; ↑: Increase; ↓: Decrease.

Algae-derived compounds and extracts with in vivo anti-cancer activity which fulfilled the inclusion criteria for this review are summarized in Table 12 (*n* = 4). These results followed the trends identified in the cell experiments, displaying a decrease in tumor proliferation due to a reduced migration capacity [220], angiogenesis capacity [221], inflammation [220,221] (downregulation of NF-κB), increased apoptosis (mainly through the upregulation of the MAPK pathway and an increase in the Bax/Bcl2 ratio) [220,221,222,223], and anti-tumor immunity (including an increase in leukocytes) [222]. Interestingly, dieckol [223], a phlorotannin, decreased cell tumor survival and restored skin tissue injuries while increasing antioxidant enzymes in skin cancer, adding to the debate of ROS as a therapeutic target. In case of Ehrlich ascites carcinoma, the methanol extracts from *Jania rubens* and *Padina pavonica* additionally displayed protective effects on kidney and liver [222]. The antiproliferative activity of *Ecklonia cava* ethanol extract can probably be attributed to its phlorotannin content [220]. Based on these results, it seems that several algae-derived extracts, fractions, and compounds hold promising potential as anti-cancer agents, not only targeting the classical hallmarks of cancer but also demonstrating higher selectivity towards cancerous cells than healthy ones, thereby minimizing undesirable side effects. Another promising novel strategy for the use of these compounds could be their use in combination with other already established chemotherapeutic agents, inducing pronounced anti-neoplastic effects with additional immunotropic effect, for improving outcomes and minimizing relapse episodes [195].

## 4. Future Directions and Research Gaps

### 4.1. Emerging Areas of Research in Algal Bioactivities

Algae have gained attention in biotechnology due to the diverse range of biologically active compounds present in algal biomass, making them suitable for many applications in the field of health and medicine, namely, in the nutraceutical industry [12]. Algal compounds have many activities that were not included in the present review, such as anti-microbial [224], anti-viral [225], anti-bacterial [226], neuroprotective [227], osteogenic potential [228], and immunomodulatory activities [229], which might also benefit human health. Additionally, some bioactive compounds act on plant and animal health (such as biological activities in aquaculture) [230,231] and were not explored in the present review either.

Algae have gained popularity in recent years as “functional foods”, in which the incorporation of algal biomass or compounds into food formulations aims to help mitigate nutritional deficiencies of an ever-growing population [232]. The COVID-19 pandemic heightened curiosity regarding natural functional foods, as individuals have actively pursued dietary components that boost their immune system [233].

Apart from their potential role as functional foods or food supplements, as algae remain largely underexploited, the array of undiscovered compounds with health benefits is yet to be elucidated; however, due to their unique structure, algae-based compounds could also serve as inspiration for the chemical synthesis of pharmaceutical drug development [234]. The use of algal compounds in combination with already established drugs (e.g., chemotherapeutic agents) is therefore an emerging and promising research field [195]. Additionally, nanotechnological perspectives are emerging for algal compounds, such as the synthesis of nanoparticles for advanced applications, including drug delivery systems. Algae-derived nanoparticles show potential for enhancing the targeted delivery of therapeutic agents, improving drug stability, and providing sustained release profiles [235].

### 4.2. Challenges and Limitations in Studying Algae for Health

Although algae consumption by humans dates back 14,000 years, the study of algae is a relatively new area, research on microalgae only emerged in the mid-20th century, and research on their bioactive compounds has only gained significant attention in recent decades [236,237]. In addition, there is a vast diversity of algae species, each with its unique biochemical composition and bioactive potential, which makes it difficult to identify and isolate the specific bioactive compounds relevant to health applications [6].

The lack of a standardization of methods for the cultivation and/or harvesting of macro- or microalgae poses a challenge, as variations in these processes can significantly impact the biochemical composition of algae and therefore affect the reproducibility of research findings. Many factors can influence algal biochemical composition, either biotic (e.g., the establishment of symbiotic relationships with other organisms) [238] or abiotic (e.g., light, temperature, nutrient availability, and carbon source) [239]. In the case of microalgae, the optimization of culture conditions largely depends on the nature of the target compound and also the specific algae species [240]. Additional challenges are limited market demand and consumer awareness, high cost associated with cultivation and processing, significant knowledge gaps, and the possibility of bioaccumulation of contaminants in algae biomass (e.g., heavy metals), which might pose a health threat [241].

After biomass collection, the extraction process poses the next challenge, as methods are not standardized. Among the main factors that affect extract composition and final bioactivity are the drying method of the biomass, the extraction method, the solvent ratio, and the extraction temperature and time [242,243,244]. Extraction parameters must be optimized depending on the targeted compounds and can also contribute to higher cost [245].

As this review showed, most of the research was conducted on algae extracts, rather than isolated bioactive compounds, which are of most interest to the nutraceutical industry, possibly because the isolation and purification of compounds is a long and costly process [246]. Extracts are complex mixtures of compounds, and often the purification of individual bioactives is counterproductive as the bioactivity of the crude extracts may be a result of synergistic activity of compounds and, therefore, higher than that of isolated compounds [247]. Thus, the isolation of bioactive compounds may be a double-edged sword, bringing an additional challenge to the application of algae compounds in practice.

The majority of the cell and in vivo research that met the inclusion criteria for this study focused on macroalgae (*n* = 116), most of them concerning extracts (*n* = 34) and fractions (*n* = 9), which showed antioxidant, anti-inflammatory, cardioprotective, gut health modulation, metabolic health promoter, and anti-cancer activity. Only one study [183] investigated the whole biomass, with metabolic health-enhancing activity. Isolated compounds were classifiable into fourteen categories: polysaccharides, phlorotannins, terpenes, carotenoids, polyphenols, phenols, indole derivates, chlorophyll catabolites, peptides, monoterpenoid lactones, hydroquinoes, disulfides, and fatty alcohol esters (Appendix A).

Antioxidant activity was linked to many compound types, including indole derivatives, phenols, phlorotannins, polyphenols, and polysaccharides. Anti-inflammatory properties were observed in compound classes such as carotenoids, disulfides, monoterpenoid lactones, phlorotannins, polyphenols, polysaccharides, and terpenes. Cardioprotective effects were specifically linked to peptides, polysaccharides, and terpenes. Carotenoid, hydroquinones, phlorotannins, polyphenols, polysaccharides, terpenes, and a combination of polysaccharides, phlorotannins, and other polyphenols showed protective benefits on the gut health, while chlorophyll catabolites, phlorotannins, polyphenols, and polysaccharides improved metabolic health. Phlorotannins, phycobiliproteins, and polysaccharides were identified as the isolated compounds responsible for the demonstrated anti-cancer effects. Among the 16 cell and in vivo studies included in this review on microalgae, the majority examined microalgae extracts (*n* = 11) which exhibited health benefits by displaying anti-inflammatory, cardioprotective, and anti-cancer activities. Only one study [189] examined the impact on metabolic control while using the entire biomass as a supplement. Isolated compounds from microalgal origin were found in three studies, whereas nonyl8-acetoxy-6-methyloctanoate [74], a fatty alcohol ester, and phycoerythrin [105], a phycobiliprotein, displayed anti-inflammatory activities. Sulfated polysaccharides showed both anti-cancer [219] and anti-inflammatory [105] activities. Isolated compounds of not-detailed algae origin were featured in eleven cell and in vivo papers, being classifiable as carotenoids (with metabolic health promoting [179] and anti-cancer activities [202,209]), phenols [45] (with antioxidant effect), phlorotannins (with antioxidant effect [43,44] and gut-health promoting [152] and anti-cancer activity [223]), polysaccharides (with cardioprotective [131] and anti-cancer activity [210]), and a mix of polysaccharide and carotenoid (with cardioprotective activity) [135]. Four groups of chemicals made up 42.4% of the isolated algae-based compounds used in the cell and in vivo studies in the current review. Polysaccharides accounted for 23.6% of the compounds, phlorotannins for 10.4%, and carotenoids and terpenes for 4.2% each, with various bioactivities, as depicted in Figure 2.

As shown throughout this review, algae-derived compounds have a great potential to exert an array of health-related bioactivities; however, their effectiveness and applicability depends on several parameters, such as digestibility and bio accessibility, as not all compounds can be metabolized by our organism and digestion might compromise the previously identified bioactivity. This is a special concern for when the whole biomass is used, where the cell wall may prevent compounds from exerting their biological function in the human body [248]. In this review, this concern was only briefly addressed in one research paper [78]. Therefore, additional studies evaluating the effectiveness of compounds are essential to strengthen the understanding of algal compounds for practical use [248].

One of the greatest challenges for the application of algae-derived compounds in human health is their introduction into the consumer market, as, if not traditionally used, these species must undergo the process of novel food approval by regulatory authorities. This is a special concern for microalgae, as from the plethora of microalgae species only a handful of strains are approved for human consumption in the EU. Currently, twenty-four species of seaweed (which did not have to go through the “novel foods” process) are considered as food within the EU in addition to five microalgae species: *Aphanizomenon flosaquae, Arthrospira platensis,* and three *Chlorella* strains (*Chlorella luteoviridis, Chlorella pyrenoidosa*, and *Chlorella vulgaris*). Only five additional (microalgal) species have been approved as food ingredients and are considered “novel food”: *Odontella aurita*, *Ulkenia* sp. (oil extract), *Tetraselmis chui*, *Haematoccocus pluvialis* (astaxanthin), and *Schizochytrium* sp. (oil extract). The application for new algal strains as “novel foods” is a timely and expensive endeavor. Therefore, companies would rather invest their research and development in species of algae that are already approved for human consumption, leaving many species and their compounds commercially unexplored and their activity in academia [249]. This is proven by the limited number of commercial products included in this review (*n* = 3): Fuco Pets HeartFight^®^ [136], of macroalgal origin, with cardioprotective activity; TetraSOD^®^ [189], from *Tetraselmis chui*, with metabolic benefits; and Gdue© [145], a brown algae extract and chromium picolinate blend with liver health-promoting ability. However, in February 2024, several algae species (21 macroalgae and 11 microalgae) were added to the EU Novel Food Catalogue, some as food ingredients and others as food supplements [250].

The limited number of microalgae studies in this review may be attributed to several factors: the higher difficulty and cost associated with cultivating microalgae compared to collecting seaweed biomass [241]; microalgae having a shorter research history than seaweeds [236]; the lower number of approved microalgae strains for consumption [249]; and the specific inclusion criteria of the review (limited to marine-origin algae, excluding in chemico studies, only including studies with results in the form of IC_50_ values in in vitro assays, and focusing on specific algal bioactivities related to human health). These inclusion criteria were chosen according to the pipeline for drug development, whereas preclinical studies with in vitro, ex vivo, and in vivo models, and clinical development with humans (Phase I, II, and III trials) are the most advanced type studies available before launching for the consumer market [251], and therefore include the algae compounds which are closest to practical applicability.

Therefore, further research is needed to address these challenges, improve methodologies, and enhance our understanding of the complex relationship between algae and human health, fostering the development of safe and effective health-promoting products which can enter the consumer market.

## 5. Conclusions

### 5.1. Summary of Key Findings on Algal Bioactivities Related to Human Health

This comprehensive review presents an overview of recent research conducted in the past five years on the impact of algae-derived compounds on human health. It provides a summary of the bioactivity and potential therapeutic properties of 92 different strains of algae, including 13 microalgae and 79 macroalgae. However, out of the 50,000 microalgae and approximately 12,000 seaweed species that have been described, many have never been studied and constitute an untapped potential that is yet to be discovered.

Out of the numerous strains discussed in this study, four macroalgal strains demonstrated multiple health-related bioactivities in in vivo models. Notably, three of the macroalgal species are classified as Phaeophiceae (brown algae) and one as chlorophyte (green algae). *Ecklonia maxima* presented antioxidant [60] and anti-inflammatory activity [113], leading to increased zebrafish larvae’s survival rate and health. *Ishige okamurae* also presented dual activity by improving both gastrointestinal (by downregulating lipogenesis, decreasing liver inflammation, and regulating lipid metabolism) [147] and metabolic health (by modulating insulin resistance and sensitivity) [187]. *Saccharina japonica* showed both anti-inflammatory (leading to increased zebrafish larvae’s survival rate and health) [95] and metabolic health-promoting activities (evidenced by a decrease in dyslipidemia, liver, and renal injury, while increasing insulin sensitivity and gut microbiota health) [184]. The polysaccharides of chlorophyte seaweed *Ulva lactuca*, also known as “sea lettuce”, were shown to possess antioxidant (by decreasing oxidative stress-induced injuries) [59] and anti-cancer activity (through various mechanisms, such as the direct killing of tumorous cells, the inhibition of angiogenesis, and a decrease in inflammation) [221].

These four stains, along with the dozens that are presented in this review, are merely an example of the unexplored potential found in algae. Anti-oxidative, anti-inflammatory, and anti-cancerous activities, along with beneficial effects for gastrointestinal and cardiovascular health, the potential of assisting with diabetes and obesity, and possibly numerous other bioactivities, could place algae in the front line of the future of natural functional ingredients, foods, and additives.

### 5.2. Implications for Future Research and Applications

Overall, several bioactivities from algae-derived compounds are shown to be effective against oxidative stress and inflammation, both of which are inextricably linked to many diseases that plague modern societies and thus play an important role in health promotion. However, before considering human consumption, the in vivo effectiveness, digestibility, and safety of these extracts must be thoroughly assessed. Other issues described in this review, such as non-standardized extractions, variable biochemical composition of algae, the scarcity of isolated bioactive compounds, and the legislation required to market these products, must be addressed for further practical applications.

While these lines are being written, hundreds of novel compounds, extracts, and whole algal biomass are being investigated as the next ingredient in food, nutraceuticals, cosmetics, and even pharmaceuticals. The big question remains: how many of these bioactive ingredients will reach the end user?

## Figures and Tables

**Figure 1 molecules-29-01900-f001:**
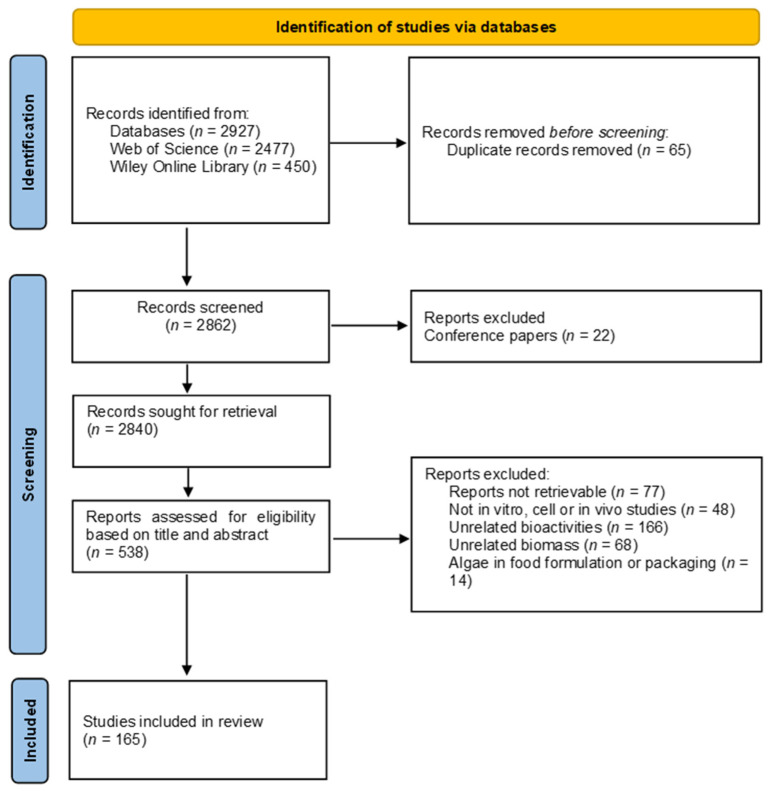
Flow diagram of the systematic review study selection process. Adapted from reference [28].

**Figure 2 molecules-29-01900-f002:**
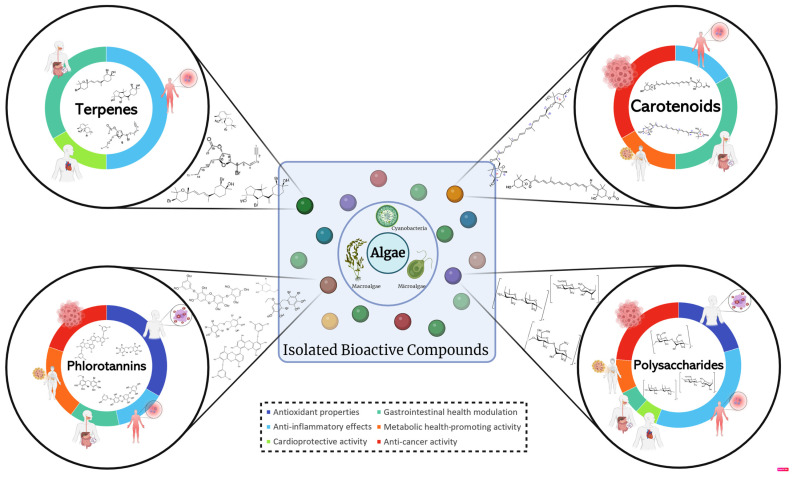
Overview of the predominant chemical classes (terpenes, carotenoids, polysaccharides, and phlorotannins) of isolated bioactive compounds from algae in cell and in vivo studies found in this review, and their respective human health-related bioactivities (antioxidant properties, anti-inflammatory effects, cardioprotective activity, gastrointestinal health modulation, metabolic health-promoting activity, and anti-cancer activity). Created with BioRender.com (accessed on 4 March 2024).

**Table 6 molecules-29-01900-t006:** In vitro studies regarding algal extracts/compounds with anti-hypertensive activities.

Algae Type	Algae Strain	Type of Analyzed Sample (Extract or Pure Compound)	ACE Inhibition (IC_50_ Values in µg/mL Unless Otherwise Stated)	References
Macroalgae	*Acrochaetium* sp.	VGGSDLQAL peptide	433.1 µM	[126]
	*Amphiroa fragilissima* (Linnaeus) J.V. Lamouroux	Ethyl acetate–methanol extracts (pigments were eliminated by means of a first extraction with hexane)	620	[73]
	*Gracilaria canaliculata* Sonder		190	
	*Gracilaria corticata* (J. Agardh)		200	
	*Gracilaria salicornia*		90	
	*Halymenia dilatata* Zanardini		230	
	*Hydropuntia edulis* (S.G.Gmelin) Gurgel & Fredericq		180	
	*Mazzaella japonica*	Protein hydrolysate YRD (sequence YRPY) VSEGLD (sequence DGL) TIMPHPR (sequence PR) GGPAT (sequence GPA) (sequence GP) SSNDYPI (sequence LKYPI) (sequence DY) SRIYNVKSNG (sequence RIY) (sequence IY) (sequence YN) (sequence VK) VDAHY (sequence VDSDVVKG) (sequence HY) YGDPDHY (sequence HY) DFGVPGHEP (sequence DFG)	262 320 µM 2.1 µM 4.1 µM 405 µM 253 µM 27.1 µM 100 µM 28 µM 2.1 µM 51 µM 13 µM 13.3 µM 26.1 µM 26.1 µM 44.7 µM	[124]
	*Padina tetrastromatica* Hauck	Ethyl acetate–methanol extracts (pigments were eliminated by means of a first extraction with hexane)	120	[73]
	*Palisada pedrochei* J.N.Norris		210	
	*Portieria hornemannii* (Lyngbye) P.C. Silva		220	
	*Sargassum horneri*	Methanol extract	440	[127]
	*Sargassum ilicifolium*	Hydrolysated protein by alcalase enzyme	1280	[128]
	*Sargassum macrocarpum*	Methanol extract (80%) Hexane fraction Chloroform fraction Ethyl acetate fraction Sargachromenol 7-methyl sargachromenol Sargaquinoic acid	380 790 180 300 0.44 mM 0.37 mM 0.14 mM	[123]
	*Spyridia filamentosa* (Wulfen) Harvey	Ethyl acetate–methanol extracts (pigments were eliminated by means of a first extraction with hexane)	240	[73]
	*Ulva intestinalis*	Unfractionated Trypsin protein hydrolysate MW < 3 kDa 3 kDa < MW < 10 kDa MW > 10 kDa	1590 1140 2190 2530	[125]

ACE: Angiotensin-converting enzyme; MW: Molecular weight.

**Table 7 molecules-29-01900-t007:** In vivo studies regarding algal extracts/compounds with cardioprotective activities.

Complication	Algae Type	Algae Species	Algal Extraction or Compound	Route of Administration	Dosage	Experimental Period	Animal Model (Age)	Induced by	*n*/Group	Outcomes and Mechanism	References
Aging	n.d.	n.d.	Low-molecular-weight fucoidan (LMWF) in combination with high-stability fucoxanthin (HSFUCO)	Oral	500 mg/kg when pure compounds or 250 mg/kg LMWF + 250 mg/kg HSFUCO	28 d	C57BL/6 mice (2 y)	-	6	↓ senescent deterioration (↓ protein expression levels of SOS1 and GRB2, ↑ GSK3, CREB, and IRS1); ameliorated malfunctions of cardiac system in aging mice (↓ cardiac fibrosis, ↑ ventricular rhythm, and ↓ action potential); better muscular function (↑ strength and ↑ muscle endurance)	[135]
Carotid atherosclerotic lesions	n.d.	n.d.	Fucoidan	Intraperitoneal injection	60 mg/kg/day	4 w	ApoE C57BL/6 mice (6 w)	HFD and high-cholesterol diet	12	↓ lipid levels (↓ TC, LDL cholesterol, and TG); ↓ unstable carotid atherosclerotic plaque formation and lipid disposition; ↑ selective Autophagy (↑ LC3II/LC3I level and ↓ p62 level); ↓ inflammasome activity (↓ IL-1β, ↓ NLRP3, ASC, and caspase-1)	[131]
Calcification of heart valves	Macroalgae	n.d.	Fucoidan—Fuco Pets HeartFight^®^ (Hi-Q Marine Biotech International Ltd., New Taipei City, Taiwan)	Oral	60 mg/kg	1.5 y	Dogs with already diagnosed heart disease	-	26	In combination with medical treatment, ↓ compensatory cardiac enlargement (decreased vertebral heart size) and recovery in echocardiographic parameters (↓ linkage of the mitral valve and tricuspid valve), showing improved overall function of ventricular contraction and relaxation	[136]
Inflammation of heart tissues	Macroalgae	*Turbinaria ornata*	Neophytadiene	Oral	50 mg/kg/day	7 d	Sprague Dawley rats (6–8 w)	LPS 10 mg/kg (intraperitoneal)	6	Improved hematological parameters (restored WBC, HCT, and PLT); improved serum markers (↓ AST); ↓ oxidative stress markers (↓ MDA; ↑ SOD); ↓ IL-6, IL-10, and PGE2 expression in heart tissue; ↓ inflammatory protein expression in heart tissue (↓ IL-1β; ↓ TNF-α; ↓ iNOS); downregulation of MAPK and NF-κB signaling pathways	[132]
Hypertension	Macroalgae	*Gracilaria tenuistipitata*	Crude neutrase hydrolysate	Oral	200 mg/kg	6 w	Spontaneously hypertensive rats	-	6	↓ systolic blood pressure	[133]
Hypertension	Microalgae	*Bellerochea malleus*	Papain protein hydrolysates	Intraperitoneal injection or oral	75 mg/kg/day (i.p.) or 400 mg/kg/day (oral)	2 w	Spontaneously hypertensive rats (11–14 w)	-	4	↓ systolic blood pressure	[134]

ASC: Apoptosis-associated speck-like protein; AST: Aspartate aminotransferase; Bax: Bcl-2-associated X protein; Bcl2: B-cell lymphoma 2; CREB: cAMP response element-binding protein; d: Days; GSK3: Glycogen synthase kinase 3; GRB2: Growth factor receptor-bound protein 2; HCT: Hematocrit; HFD: High-fat diet; HO-1: Heme oxygenase-1; ICAM: Intercellular adhesion molecule; IL: Interleukin; i.p.: intraperitoneal injection; IRS1: Insulin receptor substrate 1; LDL: Low-density lipoprotein; LC3I: Microtubule-associated proteins 1A/1B light chain 3B; MAPK: Mitogen-activated protein kinase; MDA: Malondialdehyde; MMPs: Matrix metalloproteinases; n.d.: No data or not determined; NF-κB: Nuclear factor-kappa B; NLRP3: NOD-, LRR-, and pyrin domain-containing protein 3; Nrf2: Nuclear factor erythroid 2-related factor 2; ox-LDL: Oxidized low-density lipoprotein; PGE: Prostaglandin E; PLT: Platelet count; ROS: Reactive oxygen species; SOD: Superoxide dismutase; SOS1: Son of Sevenless homolog 1; TC: Total cholesterol; TG: Triglycerides; TNF: Tumor necrosis factor; VCAM: Vascular cell adhesion molecule; w: Weeks; WBC: White blood cell count; y: Years; ↑: Increase; ↓: Decrease.

**Table 9 molecules-29-01900-t009:** In vitro studies regarding algal extracts/compounds with metabolic benefits.

Algae Type	Algae Strain	Type of Sample (Extract and/or Purified Compound)	Anti-Diabetic Activity (IC_50_ Values in µg/mL Unless Otherwise Stated)				References
			α-amylase Inhibition Activity	α-glucosidase Inhibition Activity	DPP-IV Inhibition	PTP-1B Inhibition	
Macroalgae	*Amphiroa fragilíssima* (Linnaeus) J.V. Lamouroux	Ethyl acetate–methanol extracts	870	730	50		[73]
	*Botryocladia leptopoda*	Ethanol extract	95.8	27.3			[171]
	*Caulerpa racemosa*	Ethanol extract/fraction Hexane fraction Ethyl acetate fraction	69.1 88.2 85.8	64.2 52.8 80.6			[84]
		Crude polyphenolic extract	202.5	399.1			[172]
	*Durvillaea antarctica*	Ethanol extract	473.4				[173]
		Acetone extract	466.0				
	*Fucus vesiculosus*	Conventional extraction		1.73			[174]
		Acetone (67%) extract	28.8	4.5			[165]
		Ethyl acetate fraction from extract	2.8	0.82			
	*Gracilaria bursa-pastoris*	Methanol extract (Soxhlet)	800	400			[175]
		Aqueous extract (Maceration)	900	300			
	*Gracilaria canaliculate* Sonder	Ethyl acetate–methanol extract	700	650	19		[73]
	*Gracilaria corticata* (J. Agardh)		540	530	60		
	*Gracilaria edulis* (Gmelin) Silva	Crude methanol extract Hexane fraction Chloroform fraction Ethyl acetate fraction Aqueous fraction	349.6 393.1 22.7 279.5 376.5	102.2 163.9 122.7 87.9 148.6			[176]
	*Gracilaria salicornia*	Ethyl acetate–methanol extracts	500	450	30		[73]
	*Halimeda tuna*	Ethyl acetate fraction from methanol extract	870	10			[170]
	*Halymenia dilatata* Zanardini	Ethyl acetate–methanol extracts	950	820	170		[73]
	*Hydropuntia edulis*	[“3)-4,6-O-(1-carboxyethylidene)-b-D-galp-(2SO_3_-)-(1”4)-3,6-a-LAnGalp-(2OMe)-(1”]			4.44 µM		[167]
	*Hydropuntia edulis* (S.G.Gmelin) Gurgel & Fredericq	Ethyl acetate–methanol extracts	790	630	60		[73]
	*Lessonia spicata*	Ethanol extract Acetone extract	5317.6 479.2				[173]
	*Nannochloropsis* sp.	Ethyl acetate extract	122.0	178.5			[177]
	*Padina tetrastromatica*	6-methoxy-dolabella-8(17),12-diene-10b,18-diol	180	150			[166]
		3-methoxy-dolabella-12(18)-ene-4b-ol	210	200			
		3-methoxydolabella- 10,18(19)-diene-5a,8b-diol	160	140			
		2,7-dimethoxy-14a-hydroxy-dolasta-1(15),9-diene	220	210			
		4,7-dimethoxy-9b,14a-dihydroxy-dolasta-1-ene	130	110			
	*Padina tetrastromatica* Hauck	Ethyl acetate–methanol extracts	450	400	20		[73]
	*Palisada pedrochei* J.N.Norris		610	640	80		
	*Portieria hornemannii* (Lyngbye) P.C. Silva		810	830	80		
	*Pterocladia capillacea*	Water extract (Soxhlet)		62			[169]
	*Sargassum myriocystum*	Methanol (80%) (Soxhlet)	11.5				[178]
	*Sphacelaria rigidula*	Water extract (Soxhlet)		13			[169]
	*Spyridia filamentosa* (Wulfen) Harvey	Ethyl acetate–methanol extracts	770	730	50		[73]
	*Stoechospermum marginatum*	Water extract (Soxhlet)		151			[169]
	*Turbinaria ornata*	6, 7-dihydroxy-8-methyl-3-(5′-methyloct-4′-en-1′-yl)-hexahydrocyclooct-1-en-[1, 2-c]furan-11-one (turbinafuranone A)	0.39 mM	0.34 mM		2.58 mM	[168]
		4-hydroxy-3-isopropyl-7, 8-dimethyl-6-(pentan-2′-acetate)-hexahydrocycloocta-1-en-[1, 2-c]furan 11-one (turbinafuranone B)	0.31 mM	0.27 mM		2.42 mM	
		6-acetoxy-8-ethyl-5-methoxy-3-(2′-methylhex-4′-en-1′-yl)-pentahydrocycloocta-1, 7-dien-[1, 2-c]furan-11-one (turbinafuranone C)	0.48 mM	0.44 mM		2.77 mM	

DPP-IV: Dipeptidyl peptidase-IV; PTP-1B: Protein tyrosine phosphatase 1B.

**Table 12 molecules-29-01900-t012:** In vivo studies regarding algal extracts/compounds with anti-cancerous activity.

Complication	Algae Type	Algae Species	Algal Extraction or Compound	Route of Administration	Dosage	Experimental Period	Animal Model (Age)	Induced Way	*n*/Group	Outcomes and Mechanism	Reference
Colon carcinoma	Macroalgae	*Ecklonia cava*	Ethanol extract	Oral	250, 500, and 1000 mg/mL	37 d	BALB/cKorl syngeneic mice (7 w)	4 × 10^5^ CT26 cells were injected subcutaneously	7	↓ tumor growth (↓ volume and weight); suppression of tumor proliferation; ↑ apoptosis (↑ phosphorylation of members of the MAPK signaling pathway and Bax/Bcl2 signaling pathway); ↓ migration ability of tumor cells; tumor suppressing activity (downregulation of the NF-ΚB signaling pathway)	[220]
Ehrlich ascites carcinoma	Macroalgae	*Jania rubens*	Methanol extract	i.p. injection	2.3 µg/mouse and 1.2 µg/mouse	14 d	Swiss albino mice (6–8 w)	0.25 × 10^6^ EAC cells were i.p. implanted into naïve female Swiss albino mice	11	↓ tumor growth; anti-tumor immunity (↑ immunological response in cancer; immunostimulant of the immune system); ↑ tumor apoptosis (↑ cancerous cells apoptosis, cancerous cell cycle arrest–prevention of cancer progression); ↑ leucocytes (↓ leukocytosis by tumor inoculation); ↑ organ health (restored liver function and integrity, hepaprotective role, ↓ initiation and progression of nephrocellular injury)	[222]
		*Padina pavonica*			2.5 µg/mouse and 1.3 µg/mouse				10		
Hepatoma	Macroalgae	*Ulva lactuca*	Polysaccharide	Oral	150 and 300 mg/kg	7 d	Kunming mice (6 w)	H22 cell (10^8^/mL) injection	9	Anti-tumor activity (↓ tumor weight); downregulating the expressions of PI3K/Akt and mTOR, and promoting BAX/Bcl-2 ratio; ↓ tumorigenesis (↑ p53, ↓ NF-κB, and ↑ IKKα); direct killing effect on tumor cells (↓ TRAF2/TNF-α); inhibition of tumor proliferation by inhibiting angiogenesis	[221]
Skin cancer	n.d.	n.d.	Dieckol	Gavage	30 mg/kg	25 w	Swiss albino mice (6–8 w)	DMBA	6	Improved body and liver weight; ↓ tumor incidence, volume, number, and burden; ↓ pro-inflammatory cytokines (IL-6, IL-1β, and TNF-α); ↑ antioxidant enzymes (SOD, CAT, GPx, and GSH); ↑ expression of pro-apoptotic protein (p53, Bax, and caspase-3 and -9); inhibition of the NF-ƙB pathway	[223]

Bax: Bcl-2-associated X protein; Bcl: B-cell lymphoma; CAT: Catalase; d: Days; DMBA: 7,12-Dimethylbenz[a]anthracene; EAC: Ehrlich ascites carcinoma; GPx: Glutathione peroxidase; GSH: Glutathione; IKK: IκB kinase; IL: Interleukin; MAPK: Mitogen-activated protein kinase; mTOR: Mechanistic target of rapamycin; NF-κB: Nuclear factor-kappa B; n.d.: No data; PI3K/Akt: Phosphoinositide 3-kinase/protein kinase B; SOD: Superoxide dismutase; TNF: Tumor necrosis factor; TRAF2: Tumor necrosis factor receptor-associated Factor 2; w: Weeks; ↑: Increase; ↓: Decrease.

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
