# Peer review of "The Ocean’s Pharmacy: Health Discoveries in Marine Algae"

_molecules, 2024, doi:10.3390/molecules29081900_

Round 1
Reviewer 1 Report
Comments and Suggestions for Authors
The paper deals with very interesting topic. I really enjoyed reading it. Overall, it is very well written, easily read and understood, and in the same time very informative. My special praise goes to the chapters on specific pathobiochemical pathways of certain disorders and the potential role of algae in these pathways.
Chapter 4. Future direction and research gaps is also very well written, it raises all most important questions and concerns regarding potential novel uses of the algae both as the functional food and as a drug supplement.
During my revision, only some minor concerns and question raised, namely:
Is there any specific reason why you choose a five-year period for searching the papers?
It would be much readable if the text in Tables 1, 2, 4, 5, 7, 8, 10-12 is in horizontal direction; it is very difficult to read it when some text is horizontal and some vertical
Tables 2-12 titles should be written ABOVE tables, not BELLOW
Line 300: did you mean Tables 3, 4 and 5? Not 4, 5 and 6?
Line 307: you claimed “In terms of extracts, the anti-inflammatory activity (measured by the inhibition of inflammatory-inducing enzymes – COX and LOX) was most pronounced in Gloeothece sp. [66], Gracilaria salicornia and Padina tetrastromatica Hauck [67], as these presented the lowest IC50 values compared to the other species“. The IC50 values for Gloeothece sp are 10-20 times lower comparing to other two species, and I think it would be better to emphasize this
Line 335: “Skin disorders appeared as the leading complication in cell assays“; could you rewrite this sentence? It seems that cell assays had complications in the form of skin disorders
Line 400: did you mean inhibition (instead of inhibitory) of the ACE?
Line 405-407: please rewrite the sentence, it is not clear
In chapter B. Algal compounds in weight control and metabolism regulation (no line numbers) it is written „All studies are concerning macroalgae, whereas most studies investigated these activities on extracts“; did you mean „investigated these activities OF extracts“?
In Table 11, next to the reference 196, the cell line is not written completely (MDA-MB-…..?)
Throughout the whole manuscript, authors should take care of unique symbols/grammar; for example sometimes alpha or beta are written as a word, sometimes as symbol; names of enzymes are sometimes written with capital first letter, sometimes not; line numbers are missing on few pages, and then start again from 1; number of pages are not correct.
Author Response
The paper deals with very interesting topic. I really enjoyed reading it. Overall, it is very well written, easily read and understood, and in the same time very informative. My special praise goes to the chapters on specific pathobiochemical pathways of certain disorders and the potential role of algae in these pathways. Chapter 4. Future direction and research gaps is also very well written, it raises all most important questions and concerns regarding potential novel uses of the algae both as the functional food and as a drug supplement.
We would like to thank the reviewer’s comment and appraisal.
During my revision, only some minor concerns and question raised, namely:
- Is there any specific reason why you choose a five-year period for searching the papers?
In this 5-year time frame a rather large knowledge was produced (the used search string yielded 2,927 papers), therefore it was considered a suitable time-period. Also, no reviews covering this timeframe and topic were found.
- It would be much readable if the text in Tables 1, 2, 4, 5, 7, 8, 10-12 is in horizontal direction; it is very difficult to read it when some text is horizontal and some vertical
The suggestion has been accepted and all tables are now in horizontal direction. For this reason, the “sex” column from the in vivotables had to be deleted.
- Tables 2-12 titles should be written ABOVE tables, not BELLOW
The suggestion has been accepted and all the changes have been made accordingly.
- Line 300: did you mean Tables 3, 4 and 5? Not 4, 5 and 6?
It was our mistake; changes have been made accordingly.
- Line 307: you claimed “In terms of extracts, the anti-inflammatory activity (measured by the inhibition of inflammatory-inducing enzymes – COX and LOX) was most pronounced in Gloeothece sp. [66], Gracilaria salicornia and Padina tetrastromatica Hauck [67], as these presented the lowest IC50 values compared to the other species“. The IC50 values for Gloeothece sp are 10-20 times lower comparing to other two species, and I think it would be better to emphasize this
We thank the reviewer for the suggestion; changes have been made accordingly (lines 368-371).
- Line 335: “Skin disorders appeared as the leading complication in cell assays“; could you rewrite this sentence? It seems that cell assays had complications in the form of skin disorders
Thank you for the suggestion. The sentence was rephrased as “Cell assays also focused on skin-related disorders…” (line 396).
- Line 400: did you mean inhibition (instead of inhibitory) of the ACE?
Yes. The phrasing was changed according to the reviewer’s suggestion (line 460).
- Line 405-407: please rewrite the sentence, it is not clear
Following the reviewer’s request, the sentence was rephrased and now reads: “In contrast, the hydrolysate from Ulva intestinalis exhibited the opposite behavior, where bioactivity was increased in the fraction with a molecular weight below 3 kDa [125], indicating concentration of the bioactive compound(s) in this particular fraction.” (lines 465-468).
- In chapter B. Algal compounds in weight control and metabolism regulation (no line numbers) it is written „All studies are concerning macroalgae, whereas most studies investigated these activities on extracts“; did you mean „investigated these activities OF extracts“?
Yes. We’ve changed it according to what was suggested (line 697-698).
- In Table 11, next to the reference 196, the cell line is not written completely (MDA-MB-…..?)
Changes have been made to make the cell line visible, table 11 (line 870).
- Throughout the whole manuscript, authors should take care of unique symbols/grammar; for example sometimes alpha or beta are written as a word, sometimes as symbol; names of enzymes are sometimes written with capital first letter, sometimes not; line numbers are missing on few pages, and then start again from 1; number of pages are not correct.
Thank you for your suggestions. Changes were made to the text so that Alpha and beta are always referred as symbols, and the names of enzymes are always in lowercase. Line and page numbering have been corrected.
Reviewer 2 Report
Comments and Suggestions for Authors
The authors have comprehensively reviewed an important topic regarding the emerging subset of marine natural products derived from unicellular or multi-cellular algae that suggest potentially significant health effects in six areas. In my view, this will become the go-to review for those interested in the most current work in this area.
There are some comments below that suggest minor changes in layout and also point out what appear to be some errors in labeling.
1. There is an apparent error in the line numbering in the manuscript. Following Section 3, line numbering appeared to be eliminated (following Table 8) and then renumbering began at “1” for Section 3.6. Please explain or correct.
2. The Intro does a good job of introducing the organization of the contribution verbally (6 topics, each including a broad intro and then alga discoveries, usually with 3 separate tables for each topic (in vitro, cell based, and in vivo). Nonetheless, this organization is not so obvious to me as I started to read it – a Table of Contents with the six topics and a separate summary listing of the Tables (especially since all topic do not have 3 tables) would really help guide the reader through this manuscript and ensure this review receives the attention it deserves.
3. Beginning on Line 300, it appears that Tables 3,4,5 are mistakenly referred to as Tables 4,5,6.
4. Table 7 is apparently mistakenly referred to as “in vitro” in the title of the Table, while in the text, line 422, this material in referred to as “in vivo” (line 424), which appeared to be correct.
Author Response
The authors have comprehensively reviewed an important topic regarding the emerging subset of marine natural products derived from unicellular or multi-cellular algae that suggest potentially significant health effects in six areas. In my view, this will become the go-to review for those interested in the most current work in this area.
We thank the reviewer for his/her kind words.
There are some comments below that suggest minor changes in layout and also point out what appear to be some errors in labeling.
1. There is an apparent error in the line numbering in the manuscript. Following Section 3, line numbering appeared to be eliminated (following Table 8) and then renumbering began at “1” for Section 3.6. Please explain or correct.
Thank you for your suggestion, corrections have been made accordingly.
2. The Intro does a good job of introducing the organization of the contribution verbally (6 topics, each including a broad intro and then alga discoveries, usually with 3 separate tables for each topic (in vitro, cell based, and in vivo). Nonetheless, this organization is not so obvious to me as I started to read it – a Table of Contents with the six topics and a separate summary listing of the Tables (especially since all topic do not have 3 tables) would really help guide the reader through this manuscript and ensure this review receives the attention it deserves.
Thank you for your suggestion. A table of contents has been added (lines 104-134) as well as a table of Tables and Figures (lines 135-153).
3. Beginning on Line 300, it appears that Tables 3,4,5 are mistakenly referred to as Tables 4,5,6.
Yes, thank you for noticing. In fact, the tables were mistakenly labelled, and the necessary corrections were done (lines 358-359).
4. Table 7 is apparently mistakenly referred to as “in vitro” in the title of the Table, while in the text, line 422, this material in referred to as “in vivo” (line 424), which appeared to be correct.
Yes, apologies for the confusion. Changes have been made accordingly (line 493).
Reviewer 3 Report
Comments and Suggestions for Authors
The review is well written and exhaustive.
As a chemist I would like to have at least part of the cited compounds drawn in a single figure, even only in supplementary. Would this be possible?
Author Response
The review is well written and exhaustive.
As a chemist I would like to have at least part of the cited compounds drawn in a single figure, even only in supplementary. Would this be possible?
Following the reviewer’s suggestion, a supplementary figure (Figure S1) has been created showing representative examples of some of the compounds cited, as well as their chemical classification and the algae in which they were detected.
Reviewer 4 Report
Comments and Suggestions for Authors
1. Abstract
- The authors should mention examples of identified bio-active compounds.
2. Introduction:
- The authors should mention examples of medications used to treat non communicable diseases highlighting their adverse effects.
- Please shorten the general part mainly from the line 27 to 41, instead you can expand on the algae species and their relation to non-communicable diseases.
3. Antioxidant properties
- Line 240, and 241; “Isolated compounds from Sargassum thunbergii (phenol)” like what? and also ‘Ecklonia maxima’ (polysaccharide) like what?.
4. Mechanisms of inflammation modulation
- The authors should mention examples of anti-inflammatory drugs and mention their side effects.
5. Cardiovascular part
- Lines 431:433; “Isolated compounds were isolated polysaccharides, polysaccharides in combination with carotenoids and terpenes”. Please rewrite these sentence.
- The authors should also mention most important compounds that were isolated and which ones have a role for controlling hypertension, MI, and atherosclerosis.
6. Algal compounds and their potential for improving impairment of gastrointestinal health
- Lines from 539 to 543; “These findings indicate that compounds derived from macroalgae (carotenoids, hydroquinones, oligoporphyrans, phlorotannins, polyphenols, polysaccharides, terpenes and combination from polysaccharides, phlorotannins and other polyphenols) may hold significant promise for the treatment of IBD”, you mentioned different classes. What are major ones that given the desired effect or mention they work in synergism?.
7. Obesity, diabetes, and metabolic health
- JNK; abbreviation for what. You could mention (Jun N-terminal kinase).
NF-κB; please mention the full name of abbreviation (Nuclear factor kappa-light-chain-enhancer of activated B cells)
- “(dolabellanes and dolastanes, sulfated py[1]ruvylated polysaccharide and turbinafuranones, respectively)”, are these compounds have antidiabetic action?.
- “In the case of Fucus vesicu[1]losus, the bioactive compounds were concentrated in the ethyl ace [1]tate fraction of the acetone extract, where the main compounds identified were phlorotannins”, please mention the antidiabetic action of phlorotannins and compare between it and the extract in the mentioned activity.
8. Anti-cancer activity
“Regarding breast cancer, the anti-proliferative activities of Chaetomorpha sp. and Nannochloropsis oculate extracts may be linked to the high content of dichloracetic acid, oximes, and L-a-Terpinol and terpenoids, carotenoids, polyphenolic and fatty acids in the respective extracts”. Please mention, how these classes work against cancer.
9. Figures are highly recommended to display some of the information instead of the long text.
Comments on the Quality of English Language
The authors should add a graphic abstract.
The authors should do English editing to all manuscript.
Author Response
- Abstract
- The authors should mention examples of identified bio-active compounds.
We thank the reviewer for the suggestion. Information regarding the main compound classes detected in algae has been added (lines 22-24).
- Introduction:
- The authors should mention examples of medications used to treat non communicable diseases highlighting their adverse effects.
Thank you for the suggestion. Examples of medications with known adverse effects has been added (lines 76-77; and in lines 733-736 regarding inflammation).
- Please shorten the general part mainly from the line 27 to 41, instead you can expand on the algae species and their relation to non-communicable diseases.
Thank you for the suggestion. The general part contained in the first paragraph of the introduction was summarized and more information regarding algae classification was added (lines 42-48). The relation between NCDs and algae is established between lines 55-61.
- Antioxidant properties
- Line 240, and 241; “Isolated compounds from Sargassum thunbergii (phenol)” like what? and also ‘Ecklonia maxima’(polysaccharide) like what?.
As requested by the reviewer, the information regarding compounds isolated from these species was added as follows: “Phenolic compounds extracted from Sargassum thunbergii [48], such as benzene and its derivatives (protocatechuic acid, difucol, gallic acid, and 4-hydroxybenzoic acid), cinnamic acids and their derivatives (p-Coumaric acid), flavonoids (isoquercitrin, quer-citrin, isorhamnetin, and catechin), and phlorotannins (bifuhalol, pentafuhalol A, 7-hydroxyeckol, deshydroxypentafuhalol, trifuhalol A), along with a sulfated polysaccharide from Ecklonia maxima [60], were found to reduce the production of reactive oxygen species (ROS) and repair skin damage.” (lines 295-299)
- Mechanisms of inflammation modulation
- The authors should mention examples of anti-inflammatory drugs and mention their side effects.
As suggested by the reviewer, information regarding NSAIDS was added (lines 354-356).
- Cardiovascular part
- Lines 431:433; “Isolated compounds were isolated polysaccharides, polysaccharides in combination with carotenoids and terpenes”. Please rewrite these sentence.
- The authors should also mention most important compounds that were isolated and which ones have a role for controlling hypertension, MI, and atherosclerosis.
The sentence was rephrased as suggested and now reads: “Hydrolysates [133,134], terpenes [132], a mix of a polysaccharide and a carotenoid [135] and isolated polysaccharides [131,135,136] showed favorable outcomes for hypertension [133,134], myocardial inflammation [132], aging [135], carotid atherosclerotic lesions [131] and heart valve calcification [136], respectively.” (line 488-492).
- Algal compounds and their potential for improving impairment of gastrointestinal health
- Lines from 539 to 543; “These findings indicate that compounds derived from macroalgae (carotenoids, hydroquinones, oligoporphyrans, phlorotannins, polyphenols, polysaccharides, terpenes and combination from polysaccharides, phlorotannins and other polyphenols) may hold significant promise for the treatment of IBD”, you mentioned different classes. What are major ones that given the desired effect or mention they work in synergism?.
We thank the reviewer for the comment. The fact that these compounds can display effects isolated or in combination was added to the text (line 594-601).
- Obesity, diabetes, and metabolic health
- JNK; abbreviation for what. You could mention (Jun N-terminal kinase).
The full name and abbreviation are presented in line 234, hence the abbreviations were maintained afterwards.
NF-κB; please mention the full name of abbreviation (Nuclear factor kappa-light-chain-enhancer of activated B cells).
The full name and abbreviation are presented in line 239, hence the abbreviations were maintained afterwards.
- “(dolabellanes and dolastanes, sulfated py[1]ruvylated polysaccharide and turbinafuranones, respectively)”, are these compounds have antidiabetic action?.
Yes, the action of the referred compounds is comparable to that of standard anti-diabetic compounds. This information was added in lines 703-709.
- “In the case of Fucus vesicu[1]losus, the bioactive compounds were concentrated in the ethyl ace [1]tate fraction of the acetone extract, where the main compounds identified were phlorotannins”, please mention the antidiabetic action of phlorotannins and compare between it and the extract in the mentioned activity.
Further information comparing the extract and fraction activity was added (lines 712-718) following the reviewer’s suggestion.
- Anti-cancer activity
“Regarding breast cancer, the anti-proliferative activities of Chaetomorpha sp. and Nannochloropsis oculate extracts may be linked to the high content of dichloracetic acid, oximes, and L-a-Terpinol and terpenoids, carotenoids, polyphenolic and fatty acids in the respective extracts”. Please mention, how these classes work against cancer.
The papers cited do not explain the mechanisms of action of the named compounds, only mentioning “by decreasing cell viability and inducing morphological changes in cancerous cells”, which was added (Lines 866-867).
- Figures are highly recommended to display some of the information instead of the long text.
As suggested, another figure exemplifying the main compound classes referred in the manuscript was added.
Comments on the Quality of English Language
The authors should add a graphic abstract.
A graphical abstract was provided with the initial submission.
The authors should do English editing to all manuscript.
The English language of the manuscript was thoroughly revised.
Round 2
Reviewer 4 Report
Comments and Suggestions for Authors
- Lines 111-162: Delete
Author Response
Following the reviewer's suggestion, lines 111 to 162 (table of contents added on R1) were deleted.